# Neural Mutual Information Estimation in Real Time via Pre-trained Hypernetworks

## Abstract

Measuring statistical dependency between high-dimensional random variables is fundamental to data science and machine learning. Neural mutual information (MI) estimators offer a promising avenue, but they typically require costly test-time iterative optimization for each new dataset, making them impractical for real-time applications. We present *FlashMI*, a pretrained, foundation model-like architecture that eliminates this bottleneck by directly inferring MI in a single forward pass. Pretrained on large-scale synthetic data covering diverse distributions and dependency structures, *FlashMI* learns to identify distributional patterns and predict MI directly from the input dataset. Comprehensive experiments demonstrate that *FlashMI* matches state-of-the-art neural estimators in accuracy while achieving 100× speedup, can seamlessly handle varying dimensions and sample sizes through a single unified model, and generalizes zero-shot to real-world tasks, including CLIP embedding analysis, motion trajectory modeling and robotics manipulation. By reformulating MI estimation from an optimization problem to a direct inference task, *FlashMI* establishes a foundation for real-time statistical dependency analysis.

## 1 Introduction

Understanding statistical dependencies between variables is fundamental to data science and machine learning. Quantifying how variables influence each other uncovers hidden structures and causal mechanisms that drive complex systems. Applications span a wide range of domains: in healthcare, identifying dependencies between lifestyle factors and disease risks enables personalized prevention strategies (Du et al., 2024); in autonomous driving, modeling dependencies between sensor signals and road conditions improves safety and decision making (Maanpää et al., 2025); in machine learning security, dependency analysis between generated and real samples strengthens out-of-distribution detection (Zheng et al., 2023a). As data becomes increasingly high-dimensional and heterogeneous, accurate and efficient dependency quantification is critical for reliable analysis, prediction, and decision-making.

Mutual information (MI) (Shannon, 1948) has long served as a principled measure for dependency, uniquely capturing complex nonlinear relationships for multivariate variables in interpretable units of bits. Its generality has made it a core tool in causal discovery, generative modeling and representation learning (Chen et al., 2016; Oord et al., 2018). However, computing MI from empirical samples is notoriously difficult: closed-form solutions exist only for simple distributions, and neural estimation methods (Belghazi et al., 2018; Duong & Nguyen, 2023; Franzese et al., 2023; Tschannen et al., 2019; Tsai et al., 2020; Nguyen et al., 2010) require costly gradient-based optimization for every pair of data. This makes them impractical for real-time or large-scale applications where rapid dependency assessment is essential.

In this work, we introduce *FlashMI*, a pretrained neural architecture for fast and differentiable estimation of statistical dependency between *multivariate* random variables. *FlashMI* predicts the strength of correlation between variables in a single inference step, while preserving full differentiability for seamless integration into larger computational pipelines. This capability enables decision-making systems to incorporate precise and reliable dependency awareness. Trained extensively on large-scale synthetic datasets that capture a wide range of dependency structures and data patterns, *FlashMI* acquires a rich inductive bias for perceiving diverse statistical relationships. Extensive experiments demonstrate that the model generalizes effectively from synthetic settings to complex real-world data,

Figure 1: Existing neural MI estimators e.g. MINE (Belghazi et al., 2018) (left) requires iterative gradient-based optimization to train a neural network for each new dataset. In contrast, we uses a *pre-trained* architecture to directly generate MI estimates in a single forward pass (right), eliminating per-dataset training and achieving speedup while maintaining comparable accuracy.

accurately capturing a broad spectrum of dependencies. The resulting scores are computationally efficient, robust, and interpretable, making *FlashMI* a versatile tool for applications where understanding variable relationships is critical.

Our contributions are summarized as follows:

- We introduce *FlashMI*, the first pretrained architecture for real-time estimation of statistical dependence between *multivariate* variables. *FlashMI* achieves accuracy on par with state-of-the-art neural methods without requiring gradient-based optimization, and flexibly handles variables of varying dimensionalities and sample sizes with a *single* pre-trained model.

- We propose an attentive dual-path hypernetwor-based architecture, pretrained on large-scale synthetic datasets covering diverse dependency structures. This design enables *FlashMI* to predict dependency strength in a single inference step while preserving full differentiability, and to generalize effectively to unseen real-world scenarios without task-specific finetuning.

- We comprehensively evaluate *FlashMI* on both synthetic benchmarks and real-world applications, including CLIP embedding analysis (Radford et al., 2021), motion trajectory modeling and robotics manipulation. Results demonstrate its robust performance and accurate perception of a wide spectrum of dependencies.

## 2 PROBLEM STATEMENT

**Quantifying nonlinear correlation with mutual information.** Consider the problem of quantifying statistical correlation between multivariate random variables $\mathbf{x} \in \mathbb{R}^{d_x}$ and $\mathbf{y} \in \mathbb{R}^{d_y}$. Unlike linear correlation coefficients that capture only linear relationships, mutual information (MI) provides a distribution-free measure for both linear and nonlinear correlations. Formally, MI is the Kullback-Leibler (KL) divergence between the joint distribution $p_{\mathbf{x},\mathbf{y}}$ and the product of marginals $p_{\mathbf{x}} \otimes p_{\mathbf{y}}$ (Kullback, 1997):

$$\mathbb{I}(\mathbf{x}, \mathbf{y}) = \mathrm{KL}(p_{\mathbf{x},\mathbf{y}} \| p_{\mathbf{x}} \otimes p_{\mathbf{y}}) = \int_{\mathcal{Y}} \int_{\mathcal{X}} p_{\mathbf{x},\mathbf{y}}(\mathbf{x}, \mathbf{y}) \log \left( \frac{p_{\mathbf{x},\mathbf{y}}(\mathbf{x}, \mathbf{y})}{p_{\mathbf{x}}(\mathbf{x}) p_{\mathbf{y}}(\mathbf{y})} \right) d\mathbf{x} d\mathbf{y}. \tag{1}$$

Strong correlation manifests as significant divergence between $p(\mathbf{x}, \mathbf{y})$ and $p(\mathbf{x})p(\mathbf{y})$, yielding large MI, while uncorrelated variables satisfy $p(\mathbf{x}, \mathbf{y}) \approx p(\mathbf{x})p(\mathbf{y})$, resulting in MI near zero.

While MI offers a principled correlation measure, it rarely admits closed-form solutions except for Gaussian distributions. Thus, practical applications require estimation from finite samples $\mathcal{D} = \{\mathbf{x}^i, \mathbf{y}^i\}_{i=1}^n$ drawn from $p_{\mathbf{x},\mathbf{y}}$. Recent advances have produced powerful neural estimators (Belghazi et al., 2018; Duong & Nguyen, 2023; Franzese et al., 2023; Tsai et al., 2020; Song & Ermon, 2019; Letizia et al., 2024; Tsur et al., 2023), with the most prominent leveraging the Donsker-Varadhan (DV) representation (Donsker & Varadhan, 1983):

$$\mathbb{I}(\mathbf{x}, \mathbf{y}) \coloneqq \sup_{\theta} \mathbb{E}_{p_{\mathbf{x},\mathbf{y}}}[\theta] - \log(\mathbb{E}_{p_{\mathbf{x}} \otimes p_{\mathbf{y}}}[\exp(\theta)]), \tag{2}$$

where $\theta : \mathcal{X} \times \mathcal{Y} \to \mathbb{R}$ is a critic function. Mutual Information Neural Estimation (MINE) (Belghazi et al., 2018) parameterizes $\theta$ as a neural network and approximates the supremum through gradient-based optimization. Alternative neural approaches also exist; see related works for a discussion.

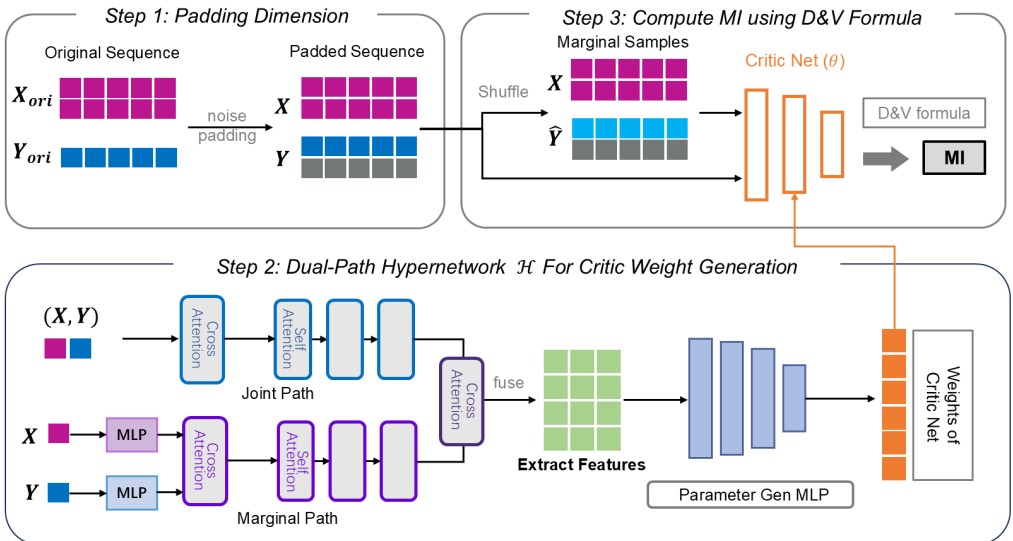

Figure 2: The *FlashMI* estimation pipeline. Step 1: We pad input dimensions with noise to ensure all variables share the same dimensionality, while allowing flexible sample sizes. Step 2: A dual-path hypernetwork $\mathcal{H}$—with joint and marginal branches—extracts features in alignment with the D-V formulation (Eq. 2). Cross-attention integrates these features, which a parameter-generation MLP then uses to produce the critic network weights. Step 3: The empirical D-V formula (Eq. 5) is applied to joint and marginal samples, with marginals obtained by index permutation, to estimate MI. This pipeline enables single-pass estimation without gradient-based optimization.

**Challenge of real-time MI estimation.** Despite their theoretical soundness, all existing neural estimators share a critical computational bottleneck: they require training a network $\theta$ from scratch for each dataset $\mathcal{D} = \{\mathbf{x}^i, \mathbf{y}^i\}_{i=1}^n$ via gradient descent:

$$\theta^{t+1} \leftarrow \theta^t - \eta \nabla_{\theta^t} \mathcal{L}(\theta^t), \quad t = 1, ..., T \tag{3}$$

where $\mathcal{L}(\theta)$ is an estimator-specific objective (e.g., the negative DV bound for MINE). Achieving accurate MI estimates typically requires thousands of gradient steps, resulting in computational complexity $\mathcal{O}(T)$. This prohibitive cost limits real-time applications such as high-frequency correlation monitoring or large-scale genomic screening. The recent InfoNet (Hu et al., 2024) addresses this inefficiency by pretraining a network to directly output optimal critic values via lookup tables, eliminating inference-time optimization. However, InfoNet is fundamentally limited to univariate inputs – extending to $d$-dimensional variables would require storing $\mathcal{O}(e^d)$ values, becoming intractable even for $d = 10$. These limitations motivate our fundamentally different approach for efficient statistical dependence measurement.

## 3 METHOD

We present *FlashMI*, a neural architecture to address the above challenge of real-time correlation estimation between multivariate random variables $\mathbf{x} \in \mathbb{R}^{d_x}$ and $\mathbf{y} \in \mathbb{R}^{d_y}$. Unlike existing neural estimators that require iterative optimization, *FlashMI* directly outputs mutual information (MI) in a single forward pass. This capability is enabled by two key innovations: (i) a dual-path attention-based hypernetwork that generates distribution-specific critic parameters directly from observed samples, which applies to datasets with varying sizes and varying dimensionality; (ii) a comprehensive pre-training strategy using synthetically generated distributions with diverse correlation structures. The former ensures that the architecture is capable of learning complex correlation structures, while the latter guarantees that the learned model is generalizable across different application domains.

## 3.1 Direct Optimal Critic Generation with Hypernetwork

Specifically, our key innovation is to reformulate MI estimation from a test-time optimization problem into a direct inference task with the aid of a hypernetwork, which generates critic function parameters rather than values conditional on observed samples. Given a dataset $\mathcal{D} = \{(\mathbf{x}^i, \mathbf{y}^i)\}_{i=1}^n$ drawn from an unknown joint distribution, *FlashMI* employs a hypernetwork $\mathcal{H} : \mathcal{D} \mapsto \Theta$ that directly outputs the complete parameter set $\theta^*$ of the optimal critic network in the Donsker-Varadhan representation (Eq. 2) via a single feedforward pass[1]:

$$\theta^* = \mathcal{H}(\mathcal{D}) = \mathcal{H}(\{(\mathbf{x}^i, \mathbf{y}^i)\}_{i=1}^n) \tag{4}$$

and leverages it to obtain an empirical MI estimation:

$$\hat{\mathbb{I}}_\theta(\mathbf{x}, \mathbf{y}) = \frac{1}{n} \sum_{i=1}^n \theta(\mathbf{x}^i, \mathbf{y}^i) - \log \left( \frac{1}{n} \sum_{j=1}^n \exp(\theta(\mathbf{x}^j, \mathbf{y}^{\pi(j)})) \right), \tag{5}$$

where $\{(\mathbf{x}^j, \mathbf{y}^{\pi(j)})\}_{j=1}^n$ denotes the marginal pairs with $\pi$ a random permutation of indices $\{1, ..., n\}$. This eliminates the iterative gradient updates required by neural MI estimators while avoiding the exponential value storage of InfoNet's lookup table approach. This architectural shift fundamentally changes the computational complexity from $\mathcal{O}(T)$ gradient steps, where $T$ is the number of optimization iterations, to $\mathcal{O}(1)$ feedforward propagation.

The hypernetwork $\mathcal{H}$ takes the form of an attentive network and consists of the following key modules:

**The joint distribution path** processes $n$ paired samples $\{(\mathbf{x}^i, \mathbf{y}^i)\}_{i=1}^n$ to extract correlation patterns inherent in $p(\mathbf{x}, \mathbf{y})$. Each sample pair is treated as a token in a sequence, enabling permutation-invariant processing through attention mechanisms. Specifically, a learnable query vector $\mathbf{q}_{\text{joint}} \in \mathbb{R}^{d_{\text{model}}}$ initiates cross-attention computation, where the concatenated samples $[\mathbf{x}^i; \mathbf{y}^i]$ serve simultaneously as keys and values. This mechanism computes attention weights $\alpha_i = \text{softmax}(\mathbf{q}_{\text{joint}}^T \mathbf{W}_K [\mathbf{x}^i; \mathbf{y}^i] / \sqrt{d_{\text{model}}})$, producing an aggregated representation that emphasizes sample pairs exhibiting strong correlations. The aggregated features subsequently pass through 16 self-attention layers ultimately producing a comprehensive encoding $\mathbf{h}_{\text{joint}} \in \mathbb{R}^{d_{\text{hidden}}}$ that characterizes the joint distribution's correlation structure.

**The marginal distribution path** processes samples from the product of marginals $p(\mathbf{x})p(\mathbf{y})$ by breaking the pairing relationship. Specifically, the samples $\{\mathbf{x}^i\}_{i=1}^n$ and $\{\mathbf{y}^j\}_{j=1}^n$ are passed through separate projection networks $f_{\mathbf{x}} : \mathbb{R}^{d_x} \to \mathbb{R}^{d_{\text{proj}}}$ and $f_{\mathbf{y}} : \mathbb{R}^{d_y} \to \mathbb{R}^{d_{\text{proj}}}$, implemented as Multi-Layer Perceptrons (MLPs) to obtain higher-dimensional representations that facilitate correlation detection. The architecture employs bidirectional cross-attention with two sets of learnable query vectors: $\mathbf{q}_{\mathbf{x} \to \mathbf{y}}$ attends from projected $\mathbf{x}$ representations to projected $\mathbf{y}$ representations (as keys and values), while $\mathbf{q}_{\mathbf{y} \to \mathbf{x}}$ performs the reverse attention. The outputs from both directions are summed element-wise, capturing symmetric independence patterns that should appear when variables lack correlation. This combined representation is then processed by 8 self-attention layers, resulting in encoding $\mathbf{h}_{\text{marginal}} \in \mathbb{R}^{d_{\text{hidden}}}$ that provides a baseline representation against which the correlation strength can be measured.

**The integration and generation module** fuses information from both distributional paths through a cross-attention mechanism that allows the joint distribution features to be modulated by marginal distribution patterns. In particular, we compute cross-attention where $\mathbf{h}_{\text{marginal}}$ serves as the query and $\mathbf{h}_{\text{joint}}$ provides both keys and values, producing a fused representation $\mathbf{h}_{\text{fused}} = \text{CrossAttention}(\mathbf{h}_{\text{marginal}}, \mathbf{h}_{\text{joint}}, \mathbf{h}_{\text{joint}})$. This asymmetric fusion ensures that correlation patterns identified in the joint path are evaluated against the independence baseline from the marginal path. The fused features are then processed by a parameter generation MLP serving as a nonlinear mapping from distributional features to critic network parameters. The MLP outputs a flattened vector $\theta \in \mathbb{R}^{|\Theta|}$ containing all parameters for a critic network, where $|\Theta| = \sum_{l=1}^L (d_l \times d_{l-1} + d_l)$ accounts for both weight matrices and bias terms across all layers.

---

[1]While primarily focusing on the D-V representation due to its simplicity, our method is also fully compatible with other variational estimators, e.g. (Song & Ermon, 2019; Letizia et al., 2024). We find that this simple setup already achieves decent performance in moderate dimensionality settings when paired with massive pre-training.

**Noise padding module** further addresses the challenge of varying input dimensions through a unified data preprocessing strategy that maintains MI while enabling consistent model architecture. For inputs with dimensions $d < D$, we pad variables with independent Gaussian noise $\mathcal{N}(0, \mathbf{I})$ to reach $D$ dimensions. This padding preserves mutual information exactly since $\mathbb{I}(\mathbf{x}, \mathbf{y}) = \mathbb{I}([\mathbf{x}; \mathbf{n}_x], [\mathbf{y}; \mathbf{n}_y])$ when noise vectors $\mathbf{n}_x$ and $\mathbf{n}_y$ are independent of all other variables, as proven in Proposition 3.

## 3.2 LARGE-SCALE PRE-TRAINING FOR UNIVERSAL CORRELATION ESTIMATION

The generalization capacity of *FlashMI* critically depends on exposure to a comprehensive spectrum of correlation structures during pre-training. For this purpose, we construct a meta-distribution $p(\mathcal{D})$ over datasets by systematically generating synthetic distributions that span diverse statistical properties. This approach draws from the principle that a model trained on sufficiently diverse synthetic data can generalize to real-world data.

**Diverse synthetic distribution generation.** In Czyż et al. (2023), a diverse set of benchmarks are constructed for comprehensive evaluation of MI estimators. Inspired by their approach, we develop a principled, fully automatic approach for constructing synthetic distributions with significantly enhanced diversity. Our data generation procedure consists of two complementary steps targeting the diversity of dependence structure and marginal patterns:

*(i) Sampling from random mixture of copulas.* The first step introduces diversity in correlation structure by sampling from mixtures of copulas with varying dependence properties. Specifically, let $c_i$ be a copula chosen from a pre-defined pool $\mathcal{C}$. We generate samples $\mathbf{x}, \mathbf{y}$ according to:

$$\mathbf{x}, \mathbf{y} \sim \sum_{i=1}^{K} \pi_i c_i, \tag{6}$$

where the parameters of each copula $c_i$ and the mixture coefficients $\pi_i$ are randomly initialized. We employ both Gaussian copulas with rich correlation structures and Student's $t$-copulas with varying tail dependencies (see Appendix A.2). According to recent vector copula theory, such copula mixture is an universal approximator for the dependence structure between $\mathbf{x}$ and $\mathbf{y}$ (Chen et al., 2025).

*(ii) Marginal transformation with random flow models.* To complement the correlation diversity, we enhance marginal pattern diversity through flow-based models (Papamakarios et al., 2021; Dinh et al., 2016). We apply two flow models $f_X : \mathbb{R}^{d_X} \to \mathbb{R}^{d_X}$, $f_Y : \mathbb{R}^{d_Y} \to \mathbb{R}^{d_Y}$ with randomly initialized parameters to transform data in each training batch:

$$\mathbf{x} \leftarrow f_X(\mathbf{x}), \qquad \mathbf{y} \leftarrow f_Y(\mathbf{y}). \tag{7}$$

These invertible transformations preserve mutual information while introducing complex marginal patterns, as $\mathbb{I}(\mathbf{x}, \mathbf{y}) = \mathbb{I}(f_X(\mathbf{x}), f_Y(\mathbf{y}))$ for any bijective function. Additionally, we apply a differentiable copula transformation using the softrank function (Blondel et al., 2020), which maps each marginal distribution to approximately uniform $[0, 1]$ while preserving the correlation structure. This normalization enables the model to focus on learning essential correlation patterns rather than adapting to varying value spans that is irrelevant to the true dependence structure.

**Overall learning objective.** With comprehensive data generation, the hypernetwork ($\mathcal{H}$) parameters are optimized through a meta-learning objective that maximizes the expected accuracy of MI estimation in the distribution of training datasets. Formally, we minimize[2]:

$$\mathcal{L}(\mathcal{H}) = -\mathbb{E}_{\mathcal{D} \sim p(\mathcal{D})} \left[ \hat{\mathbb{I}}_{\mathcal{H}(\mathcal{D})}(\mathbf{x}_{\mathcal{D}}, \mathbf{y}_{\mathcal{D}}) \right], \tag{8}$$

where $p(\mathcal{D})$ represents the meta-distribution over datasets induced by our synthetic generation process, $\mathcal{H}(\mathcal{D})$ outputs the critic network parameters for dataset $\mathcal{D}$, and $\hat{\mathbb{I}}_\theta(\mathbf{x}_{\mathcal{D}}, \mathbf{y}_{\mathcal{D}})$ is the empirical MI estimate using critic parameters $\theta$ as in Eq. 5. This objective encourages the hypernetwork to learn a mapping from distributional patterns to optimal critic parameters, effectively distilling the solution to thousands of MI estimation problems into a single model.

Proposition 1 establishes that under mild conditions, the above learning objective yields a *consistent* estimate to the ground truth MI for all $\mathcal{D}$ such that $p(\mathcal{D}) > 0$, thereby converging to optimal critic $\theta^*$.

---

[2]Theoretically, optimizing D-V representation suffers from a potential high-variance (McAllester & Stratos, 2020; Song & Ermon, 2019) and a bias due to suboptimal sampling (Letizia et al., 2024). These issues are less significant in our pretraining as (a) we can generate infinitely many samples; and (b) we use a large batch size.

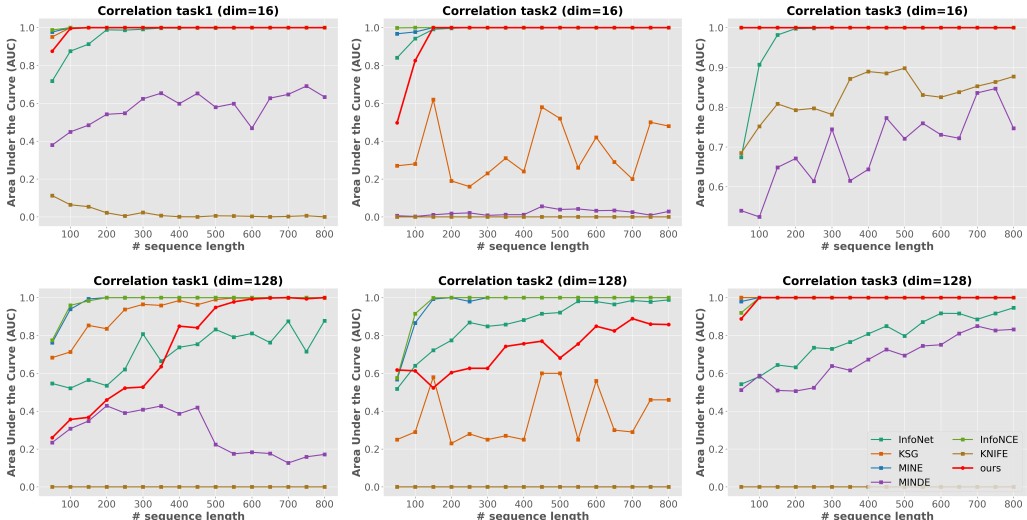

Figure 3: Independence testing under three types of data correlations. Each curve depicts the area under the curve (AUC) of the receiver operating characteristic (ROC) with respect to sequence length $n$. Seven MI estimators are compared: *FlashMI*, InfoNet, KSG, MINE, MINDE, InfoNCE and KNIFE. *FlashMI* uses 5-sliced MI with 32 slices, while InfoNet adopts 1-sliced MI with 128 slices.

## 4 EXPERIMENTS

### 4.1 SETUPS

**Slicing**. *FlashMI* is designed for data with dimensionality up to $D = 10$ per each random variable. For high-dimensional inputs where $d > D$, we employ $k$-sliced mutual information (Goldfeld et al., 2022), which projects the data onto $k$-dimensional random subspaces and averages the MI estimates across projections. Specifically, we compute $\hat{I}_{\text{k-sliced}}(\mathbf{x}, \mathbf{y}) = \frac{1}{S} \sum_{i=1}^{S} \hat{I}(\mathbf{P}_i \mathbf{x}, \mathbf{P}'_i \mathbf{y})$ where $\mathbf{P}_i, \mathbf{P}'_i \in \mathbb{R}^{k \times d}$ are random projection matrices. This approach preserves substantially more correlation structure than 1-dimensional slicing used in previous work (Hu et al., 2024), with theoretical guarantees that $I_{\text{k-sliced}} \to I(\mathbf{x}, \mathbf{y})$ as $k \to \infty$ under mild regularity conditions. For other neural methods, we do not employ slicing, as this will involve training $S$ different neural networks for $S$ slicing directions, being computationally prohibitive for even moderate slicing directions (e.g. $S = 10$).

**Baselines**. We mainly consider seven representative MI estimation methods: KSG (Kraskov et al., 2004), which uses k-nearest neighbors for entropy computation; KDE (Silverman, 2018), which employs kernel density estimation to model distributions; KNIFE (Pichler et al., 2022), which uses KDE with learnable parameters in MI estimate, and neural methods MINE (Belghazi et al., 2018), InfoNCE (Oord et al., 2018), and MINDE (Franzese et al., 2023), all of which require training a network from scratch for each new distribution. We also compare to the pretrained InfoNet (Hu et al., 2024) model whenever appropriate. InfoNet (Hu et al., 2024) is restricted to one dimension, thus requiring using a slicing technique for dimension beyond $D = 1$.

Other details including optimizer, training details and architecture can be found in Appendix A.4.

### 4.2 RESULTS

**High-dimensional independence testing.** We first evaluate *FlashMI* on its ability to accurately discriminate varying levels of statistical dependency between pairs of random variables in high-dimensional settings. Following the setup in (Goldfeld et al., 2022), we consider multiple correlation types. For each type, we generate two populations of paired variables: one with no statistical dependence and another with non-trivial dependence. The goal is to evaluate how well the estimated dependency scores separate the two populations. Performance is measured using the area under the precision-recall curve (AUC). All results are the average collected from 10 independent trials.

Table 1: Performance comparison of MI estimators on six benchmark tasks from (Czyż et al., 2023) with known ground truth. Each estimate represents the average over 10 random seeds with $N = 5000$ samples per task. Task notation indicates distribution type (Mn=Multivariate normal, St=Student-$t$, Asinh=Arc sinh, Uniform=correlated uniform, Hc=Half cube) and corresponding parameters, with the first two digits indicating dimensionality of $\mathbf{x}$ and $\mathbf{y}$ respectively. Methods are color-coded: neural-based methods in green and non-neural methods in blue. **Bold** indicates closest to ground truth, while underlined values show second-best estimates. The rightmost column shows computational time in seconds, highlighting *FlashMI*'s superior efficiency while maintaining estimation accuracy.

| Method* | Tasks | | | | | | Time (s) |
| | Mn-dense 5-5-0.5 | Spiral 3-3-2-2.0 | Asinh@St 5-5-2 | St 3-3-3 | Uniform 3-3-2-2.0 | Hc@Mn 5-5-2 | |
| --- | --- | --- | --- | --- | --- | --- | --- |
| *GT* | 0.59 | 1.02 | 0.45 | 0.18 | 1.02 | 1.02 | – |
| KSG | 0.54 | 0.75 | 0.25 | 0.07 | 0.79 | 0.58 | 0.13 |
| KDE | 1.59 | 2.87 | 2.43 | 2.36 | 1.17 | 2.23 | 2.04 |
| MINE | **0.60** | **1.00** | 0.53 | 0.21 | **1.03** | 1.06 | 25.9 |
| MINE-5s | **0.60** | 0.90 | 0.33 | 0.15 | 0.93 | 1.06 | 4.92 |
| MINDE | **0.58** | 0.92 | **0.43** | 0.36 | 0.89 | **1.01** | 34.2 |
| InfoNCE | 0.56 | 0.98 | 0.49 | **0.18** | 0.97 | 1.03 | 67.6 |
| KNIFE | 0.93 | 0.10 | 0.66 | 0.50 | 0.07 | 0.92 | 67.6 |
| *FlashMI* | **0.60** | 0.89 | 0.41 | 0.21 | 0.93 | 0.96 | **0.09** |

*We exclude InfoNet on this task, as InfoNet cannot output exact MI for data beyond 1D.

As shown in Fig. 3, *FlashMI* consistently demonstrates strong test power in assessing statistical dependence in high-dimensional settings, particularly when the sample size exceeds 400 (except for the case correlation type 2 at 128D). In these regimes, our method's performance is on par with alternative neural methods such as MINE, InfoNCE and MINDE, despite requiring no gradient-based optimization at all. Compared to InfoNet, *FlashMI* achieves comparable or superior performance in most cases but falls short in the case correlation type 2 at 128D, which may be due to the relatively smaller number of slices used ($S = 32$ in *FlashMI* vs $S = 128$ in InfoNet).

**Sanity check on STOA benchmark with known MI.** We next consider the benchmarks proposed in (Czyż et al., 2023), where select six representative tasks with analytically derived ground-truth MI. The test distributions exhibit diverse statistical patterns, ranging from Spiral transformation to heavy-tailed Student-$t$ distributions. We generate 5,000 samples for each task.

As presented in Table 1, the MI values predicted by *FlashMI* closely align with the ground-truth MI across all tasks. In all tasks, *FlashMI* achieves comparable accuracy to the best neural-based baselines, such as MINE or MINDE, while being approximately 300 times faster. Notably, *FlashMI* demonstrates robust performance on the challenging Student's $t$-distributions and Spiral transformation, which typically pose difficulties for existing estimators. Fig. 6a further visualizes the average estimation accuracy and runtime of each method, confirming that *FlashMI* offers a substantially more favorable trade-off between efficiency and accuracy than existing methods.

**Estimating mutual information for CLIP-encoded image-text representations.** The CLIP model (Radford et al., 2021) encodes images and text into a shared feature space, enabling robust cross-modal understanding by measuring similarity. Here, we assess the correlation between images and their corresponding text annotations by estimating MI between their latent representations, independently encoded by the pre-trained CLIP model.

We utilize the COCO Captions dataset (Chen et al., 2015), selecting 33,000 image-caption pairs and encoding them into 512-dimensional feature vectors using CLIP. By systematically introducing Gaussian noise to the data, we create conditions where MI naturally decreases. Our objective is to evaluate whether different MI estimators can effectively detect these changes with high sensitivity – a spirit similar to the self-consistency test in (Song & Ermon, 2019). For each noise level, we conduct 20 experiments. We report both the mean and the standard derivation.

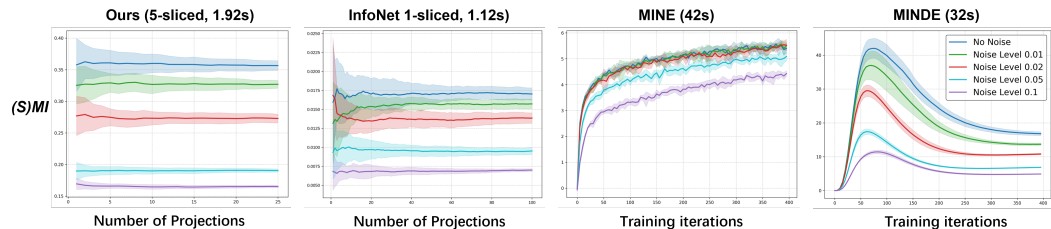

Figure 4: Comparison of MI estimation methods on 512-dimensional CLIP-encoded image-text representations across five noise levels. The light-colored areas indicate error bounds from 20 repeated experiments. **(Left to right)** *FlashMI* with 5-sliced MI using $S = 25$ random projections; InfoNet with 1-sliced MI using more projections (up to $S = 128$); MINE and MINDE estimating original MI via gradient-based optimization. *FlashMI* demonstrates superior noise level discrimination with clearly separated error bounds, while maintaining significantly faster computation time (noted in parentheses) compared to neural-based alternatives. We note that we do not employ slicing technique for MINE and MINDE, as this requires training a large number of networks for all slicing directions.

Our results in Figure 4 demonstrate that *FlashMI* achieves the strongest performance in detecting noise fluctuations, yielding clearly separated error bounds across different noise levels and hence high sensitivity w.r.t the dependence strengths, while being substantially more efficient than alternative approaches. InfoNet (Hu et al., 2024) is the only method comparable in efficiency, but its accuracy is incomparable with our method due to its reliance on 1-slicing, which discards a large amount of information despite using more slicing directions.

**Real-World Motion Trajectory Modeling** To assess *FlashMI*'s generalization ability to accurately capture complex real-world relationships, we utilize the PointOdyssey dataset (Zheng et al., 2023b), which contains multi-dimensional ground-truth motion trajectories of points on objects across video frames. In this task, a reference point $P^*$ is selected, and we estimate the mutual information $\mathbb{I}(\text{trajectory}(P^*), \text{trajectory}(P))$ between its trajectory and those of all other points $P$ in the video. Since points on the same object $O$ typically exhibit stronger spatial correlations in their trajectories than those on different objects, the following relationship is expected (where $t$ is a threshold):

$$\begin{cases} \mathbb{I}(\text{trajectory}(P^*), \text{trajectory}(P)) > t & \text{if } P^*, P \in O \\ \mathbb{I}(\text{trajectory}(P^*), \text{trajectory}(P)) \leq t & \text{if } P^* \in O, \ P \notin O \end{cases}$$

An accurate mutual information estimator should correctly reflect this relationship. Figure 5 visualizes the MI estimates between the reference point and all other points. As shown, *FlashMI* successfully identifies points that belong to the same object by flagging a high MI value, demonstrating its effectiveness in modeling complex spatial dependencies in real-world motion data. Remarkably, *FlashMI* delivers results for all point pairs within only a few seconds.

We further evaluate *FlashMI* on 12 objects sampled from 6 different videos, where we segment video objects using MI estimators. A visualization of these objects can be found in Fig. 9 and Fig. 10 in the appendix. Each method is assessed by comparing the estimated pointwise correlations with the ground-truth segmentation masks. As shown in Fig. 6b, on this out-of-distribution dataset, *FlashMI* achieves competitive segmentation accuracy while being orders more efficient.

**Robotic Manipulation Concept Discovery** To fully demonstrate our *FlashMI*'s strong potential in complex, real-world systems, we further apply our method to a robotic manipulation concept discovery task. The goal is to identify *key states*–critical moments in a trajectory $\tau^i = \{s_t^i\}_{t=1}^T$ that carry strong physical significance (e.g., "peg aligned with hole"). Identifying such key states has been shown to significantly improve robotics policy training (Zhou & Yang, 2024). Following (Zhou & Yang, 2024), we extract key states by maximizing the mutual information $I(s_{t_k}; s_{t_k-\Delta t})$ between the key state $s_{t_k}$ and its prior states $s_{t_k-\Delta t}$ ($\Delta t > 0$), where each $s_t \in \mathbb{R}^Q$ represent an environmental observation.

We train manipulation policies using key concepts extracted from different MI estimators and compare the success rates (SR) of the resulting policies. We consider three manipulation tasks from ManiSkill 2 (Gu et al., 2023)—Pick Cube, Stack Cube, and Peg Insertion. As summarized in Table 2, concepts

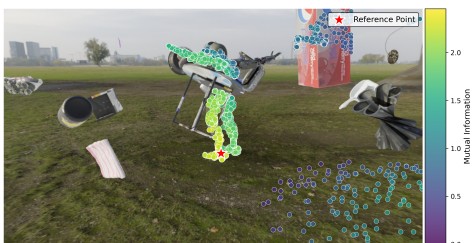
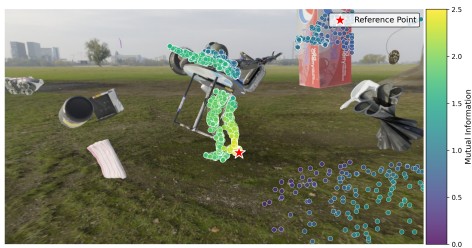

(a) Estimated Mutual Information between a reference point in object 1 and other points.

(b) Estimated Mutual Information between a reference point in object 2 and other points.

Figure 5: Application of *FlashMI* on the PointOdyssey dataset (Zheng et al., 2023b), which contains multi-dimensional motion trajectories of points on objects across video frames. We compute the mutual information $\mathbb{I}(P^*, P)$ between a reference point $P^*$ (marked with $\star$ in the figure) and every other point in the video, resulting in approximately $n \approx 4 \times 10^3$ MI terms—an order of magnitude entirely beyond the efficiency limits of existing neural estimators. As expected, our method detects substantially higher $\mathbb{I}(P^*, P)$ values for points on the same object than those on different objects.

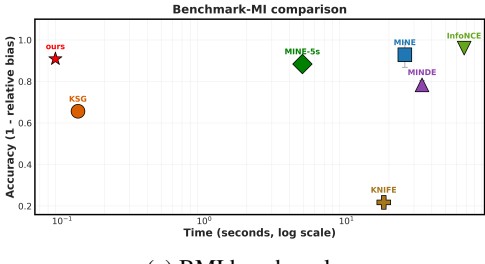
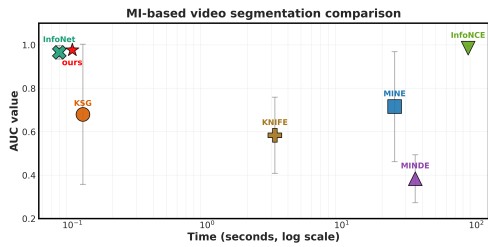

(a) BMI benchmark.

(b) Video object detection.

Figure 6: Comparing the estimation accuracy and computationally efficiency of different methods. (a) BMI task (Czyż et al., 2023), where we assess the averaged estimation accuracy across 6 tasks. Accuracy is defined as $\mathrm{accuracy} = 1 - |\hat{\mathbb{I}} - \mathbb{I}|/\mathbb{I}$. Results are collected using 10 runs. (b) Video object detection, where we compare the AUC under the precision–recall curve in object segmentation.

derived from *FlashMI* substantially outperform those from other methods in terms of success rate under equal or even reduced time budgets, highlighting that our model reliably estimates MI on unseen, complex real-world data and serves as a versatile toolbox for modern machine learning applications.

## 5 RELATED WORK

**Neural mutual information estimators**. A series of powerful, neural network-based methods have been developed for MI estimation. The most prominent among these is the MINE estimator (Belghazi et al., 2018), which builds on the Donsker–Varadhan representation. Other approaches rely on density-ratio estimation (Letizia et al., 2024; Gutmann & Hyvärinen, 2010), direct density modeling (Song & Ermon, 2019), score function estimation (Franzese et al., 2023), or leverage normalizing flows (Duong & Nguyen, 2023) and autoencoders (Gowri et al., 2024) to construct MI estimates. Despite methodological differences, these methods are primarily designed to improve estimation accuracy, and the computational overhead associated with network training is often overlooked in real-world deployment. In contrast, our work targets the orthogonal dimension of computational efficiency, replacing costly iterative optimization with a lightweight forward pass at inference time.

**Efficient computation of mutual information**. Various approaches have been developed to accelerate MI computation, each with different trade-offs. Non-parametric methods (Kraskov et al., 2004; Moon et al., 1995; Silverman, 2018) offer training-free efficiency but typically lack the capacity to capture complex dependencies in high-dimensional data. Copula-based approaches (Keziou & Regnault, 2016; Safaai et al., 2018; Purkayastha & Song, 2024; Zeng et al., 2018) balance efficiency with accuracy by assuming data follows a known copula family (e.g., Gaussian copula), but this

Table 2: Comparison of policy success rates trained by key states extracted via different MI estimators across three robotic tasks. MINE-100 means training MINE with 100 iterations. MI estimation is conducted on variables with 100 dimensions and a sample size of 100.

| Tasks | Pick Cube | | Stack Cube | | Peg Insertion | |
|---|---|---|---|---|---|---|
| | Seen | Unseen | Seen | Unseen | Seen | Unseen |
| No MI Loss | 66.0 | 60.0 | 67.4 | **41.0** | 38.6 | 9.3 |
| MINE-100 | 86.4 | 81.0 | 68.0 | 37.0 | 55.0 | 13.5 |
| MINE-1000 | 81.2 | 81.0 | 61.2 | 37.0 | 65.4 | 17.8 |
| InfoNet | 91.0 | 76.0 | 63.0 | 27.0 | 46.4 | 9.8 |
| *FlashMI* (Ours) | **94.2** | **82.0** | **68.2** | 37.0 | **72.4** | **18.3** |

assumption limits their applicability to general distributions. The recent InfoNet (Hu et al., 2024) enables fast MI estimation through neural network pretraining – a concept related to our work. However, InfoNet is restricted to scalar inputs due to its lookup table designs and limited pretraining, whereas our method supports multivariate random variables with varying dimensionalities. Our experimental comparisons and in-depth studies directly highlight these advantages in handling multivariate data.

**Foundation models for statistical analysis**. Recent advances in large-scale pretrained models have enabled direct inference on raw data without gradient-based optimization at test time. For instance, LLM-based frameworks (Requeima et al., 2024) leverages large language models to perform one-dimensional classification and regression tasks through direct inference, while the work (Siddiqui et al., 2025) investigates the functional approximation capabilities of such models for scalar prediction tasks. More recently, the work (Hollmann et al., 2025) introduced a foundation model specifically designed for univariate prediction with small tabular datasets. While these works provide valuable insights, they focus primarily on *one-way* prediction of *scalar* targets. In contrast, our work addresses the more challenging problem of quantifying *mutual* dependence between two multivariate random variables – vectors rather than scalars. This setting presents unique challenges in both architecture design and pretraining methodology, requiring our novel dual-path attention architecture and diverse synthetic distribution generation for effective generalization.

## 6 DISCUSSION

In this work, we introduce *FlashMI*, a pretrained neural architecture for MI estimation with a novel hypernetwork architecture and massive simulation-based pretraining. By eliminating iterative optimization, *FlashMI* matches state-of-the-art neural methods in accuracy while being orders faster in execution. Our validation across both synthetic distributions and complex real-world datasets confirms *FlashMI*'s strong generalization capabilities. This accuracy-efficiency combination enables new applications of MI-based methods in large-scale machine learning and real-time business and sensory data analysis. Future work will focus on enriching pretraining data diversity to enhance zero-shot generalization while preserving our key advantage in eliminating test-time optimization.

## ETHICS STATEMENT

This work strictly adheres to the ICLR Code of Ethics. This work uses only synthetic data for pre-training and publicly available, license-compliant datasets for evaluation, with all privacy-sensitive information removed. While *FlashMI* enables efficient mutual information estimation, it may be applied to sensitive data; users should ensure compliance with relevant data protection regulations and obtain proper consent where required. We caution against misuse for unauthorized surveillance or inference of personal attributes, and recommend responsible deployment with bias analysis in downstream applications.

## REPRODUCIBILITY STATEMENT

All details required to reproduce our method and the experiments results are provided in the main text and the appendix. The full details of neural architecture, batch size, optimizer are provided in A.2. The details of synthetic data generation for pretraining is provided in A.2. Full details of the high-dimensional independence test is in A.3, and the details of BMI benchmark can be found in Czyż et al. (2023). Other details, such as baseline setups and the computing resource is described under the 'setup' subsection in the experiment section. Source code will be provided upon request.

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

## A TECHNICAL APPENDIX

### A.1 THEORETICAL FOUNDATIONS

**Proposition 1** (Consistency of the estimator w.r.t sample size $n$). *Let* p *be a probability measure over datasets* $\mathcal{D}$. *For each* $\mathcal{D}$, *let* $(\mathbf{x}_{\mathcal{D}}, \mathbf{y}_{\mathcal{D}}) \sim p_{\mathbf{x}_{\mathcal{D}}, \mathbf{y}_{\mathcal{D}}}$ *with* $p_{\mathbf{x}_{\mathcal{D}}, \mathbf{y}_{\mathcal{D}}} \ll p_{\mathbf{x}_{\mathcal{D}}} \cdot p_{\mathbf{y}_{\mathcal{D}}}$ *and* $I(\mathbf{x}_{\mathcal{D}}; \mathbf{y}_{\mathcal{D}}) < \infty$. *For any admissible critic* $\theta \in \Theta$, *define*

$$\hat{I}_{\theta}(\mathbf{x}_{\mathcal{D}}; \mathbf{y}_{\mathcal{D}}) := \mathbb{E}_{p_{\mathbf{x}_{\mathcal{D}}, \mathbf{y}_{\mathcal{D}}}}[\theta(\mathbf{x}_{\mathcal{D}}, \mathbf{y}_{\mathcal{D}})] - \log \mathbb{E}_{p_{\mathbf{x}_{\mathcal{D}}} \cdot p_{\mathbf{y}_{\mathcal{D}}}}[e^{\theta(\mathbf{x}_{\mathcal{D}}, \mathbf{y}_{\mathcal{D}})}].$$

*and let* $\hat{I}_{\theta}^{n}(\mathbf{x}_{\mathcal{D}}; \mathbf{y}_{\mathcal{D}})$ *be its empirical estimate with* $n$ *sample.*

*Assume:*

*(i)* (**Per-$\mathcal{D}$ attainment**) *For* p*-a.e.* $\mathcal{D}$, *there exists* $\theta_{\mathcal{D}}^{\star} \in \Theta$ *attaining the supremum:* $I(\mathbf{x}_{\mathcal{D}}; \mathbf{y}_{\mathcal{D}}) = \sup_{\theta \in \Theta} \hat{I}_{\theta}(\mathbf{x}_{\mathcal{D}}; \mathbf{y}_{\mathcal{D}})$.

*(ii)* (**Selector universality**) *The hypernetwork class can realize any measurable selector* $\mathcal{H} : \mathcal{D} \mapsto \Theta$.

*(ii)* (**Population integrability**) *The expectation* $\mathbb{E}_{\mathcal{D}}[I(\mathbf{x}_{\mathcal{D}}, \mathbf{y}_{\mathcal{D}})]$ *satisfies* $\mathbb{E}_{\mathcal{D}}[I(\mathbf{x}_{\mathcal{D}}, \mathbf{y}_{\mathcal{D}})] < \infty$

*Define* $J(\mathcal{H}) := \mathbb{E}_{\mathcal{D}}[\hat{I}_{\mathcal{H}(\mathcal{D})}(\mathbf{x}_{\mathcal{D}}; \mathbf{y}_{\mathcal{D}})]$ *and let* $J^{n}(\mathcal{H}) := \mathbb{E}_{\mathcal{D}}[\hat{I}_{\mathcal{H}(\mathcal{D})}^{n}(\mathbf{x}_{\mathcal{D}}; \mathbf{y}_{\mathcal{D}})]$ *be its finite sample estimate with* $n$ *samples. Let* $\mathcal{H}^{\star} \in \arg\max_{\mathcal{H}} J^{n}(\mathcal{H})$. *Then the estimator*

$$\hat{I}_{\mathcal{H}^{\star}(\mathcal{D})}(\mathbf{x}_{\mathcal{D}}; \mathbf{y}_{\mathcal{D}})$$

*is a consistent estimate to* $I(\mathbf{x}_{\mathcal{D}}; \mathbf{y}_{\mathcal{D}})$ *p-a.e.* $\mathcal{D}$.

*Proof.* We begin with the following identity:

$$\sup_{\mathcal{H}} J^{n}(\mathcal{H}) = \sup_{\mathcal{H}} \mathbb{E}[\hat{I}_{\mathcal{H}(\mathcal{D})}^{n}(\mathbf{x}_{\mathcal{D}}; \mathbf{y}_{\mathcal{D}})] = \mathbb{E}[\sup_{\theta} \hat{I}_{\theta}^{n}(\mathbf{x}_{\mathcal{D}}; \mathbf{y}_{\mathcal{D}})]$$

where the second quality comes from the fact that the hypernetwork $\mathcal{H}$ is a universal selector for $\mathcal{D} \to \Theta$, so that the supremum for each $\hat{I}_{\mathcal{H}(\mathcal{D})}^{n}(\mathbf{x}_{\mathcal{D}}; \mathbf{y}_{\mathcal{D}})$ is reachable.

This suggests that for the optimal $\mathcal{H}^{*} = \arg\max J^{n}(\mathcal{H})$, we have

$$\mathcal{H}^{*}(\mathcal{D}) = \sup_{\theta} I_{\theta}^{n}(\mathbf{x}_{\mathcal{D}}; \mathbf{y}_{D})$$

According to (Belghazi et al., 2018), the estimator $\hat{I}(\mathbf{x}_{\mathcal{D}}, \mathbf{y}_{\mathcal{D}}) = \sup_{\theta} \hat{I}_{\theta}^{n}(\mathbf{x}_{\mathcal{D}}, \mathbf{y}_{\mathcal{D}})$ itself is a consistent estimate of $I(\mathbf{x}_{\mathcal{D}}, \mathbf{y}_{\mathcal{D}})$. This suggests that for each $\mathcal{D}$ and every $\epsilon > 0$, there exists $n(\mathcal{D}) \in \mathbb{N}$, such that

$$\left| I(\mathbf{x}_{\mathcal{D}}, \mathbf{y}_{\mathcal{D}}) - \sup_{\theta} \hat{I}_{\theta}^{n}(\mathbf{x}_{\mathcal{D}}, \mathbf{y}_{\mathcal{D}}) \right| \leq \epsilon, \quad \forall n \geq n(\mathcal{D}), \ a.s.$$

By taking $n' = \sup_{\mathcal{D}} n(\mathcal{D})$, substituting $\sup_{\theta} I_{\theta}^{n}(\mathbf{x}_{\mathcal{D}}; \mathbf{y}_{D}) = \mathcal{H}^{*}(\mathcal{D})$, we have that for every $\epsilon > 0$,

$$\left| I(\mathbf{x}_{\mathcal{D}}, \mathbf{y}_{\mathcal{D}}) - \hat{I}_{\mathcal{H}_{\mathcal{D}}^{*}}^{n'}(\mathbf{x}_{\mathcal{D}}, \mathbf{y}_{\mathcal{D}}) \right| \leq \epsilon, \quad \forall \mathcal{D}, \ \forall n \geq n', \ a.s.$$

which completes the proof. $\square$

**Proposition 2** (Positive Definiteness of Generated Covariance Matrix). *The covariance matrix constructed by Algorithm 2 is positive definite almost surely, with controllable condition number through the rank parameter* $m$.

*Proof.* We construct $\mathbf{\Sigma} = \mathbf{W}\mathbf{W}^{\top} + \mathbf{D}$ where $\mathbf{W} \in \mathbb{R}^{d \times m}$ with $W_{ij} \sim \mathcal{N}(0, 1)$ and $\mathbf{D} = \text{diag}(d_1, \ldots, d_d)$ with $d_i \sim \text{Uniform}(0, 1)$.

For any nonzero $\mathbf{x} \in \mathbb{R}^{d}$:

$$\mathbf{x}^{\top} \mathbf{\Sigma} \mathbf{x} = \underbrace{||\mathbf{W}^{\top}\mathbf{x}||_2^2}_{\geq 0} + \underbrace{\sum_{i=1}^{d} d_i x_i^2}_{>0 \text{ a.s.}} > 0$$

The eigenvalues satisfy $\lambda_{\min}(\boldsymbol{\Sigma}) \geq \min_i d_i > 0$ and $\lambda_{\max}(\boldsymbol{\Sigma}) \leq ||\mathbf{W}||_F^2 + \max_i d_i$. The expected condition number scales as $\mathcal{O}(m)$, allowing control over numerical stability. $\qquad\square$

**Proposition 3** (Invariance of MI under Noise Padding). *Let $(X, Y)$ be random variables with $X \in \mathbb{R}^{d_x}$, $Y \in \mathbb{R}^{d_y}$. For any independent noise variables $\epsilon_X \perp \epsilon_Y \perp (X, Y)$ of arbitrary dimensions, defining $X' = [X, \epsilon_X]$ and $Y' = [Y, \epsilon_Y]$:*

$$I(X'; Y') = I(X; Y)$$

*This invariance holds for any MI estimator, including the DV representation used in FlashMI.*

*Proof.* Since $\epsilon_X \perp \epsilon_Y \perp (X, Y)$, the joint and marginal densities factor as:

$$p(x', y') = p(x, y) \cdot p(\epsilon_x) \cdot p(\epsilon_y) \tag{9}$$

$$p(x') = p(x) \cdot p(\epsilon_x), \quad p(y') = p(y) \cdot p(\epsilon_y) \tag{10}$$

Therefore, the density ratio is preserved:

$$\frac{p(x', y')}{p(x')p(y')} = \frac{p(x, y)}{p(x)p(y)}$$

For the DV representation specifically:

$$I(X'; Y') = \sup_{\theta'} \mathbb{E}_{p(x',y')}[\theta'] - \log \mathbb{E}_{p(x')\otimes p(y')}[e^{\theta'}] \tag{11}$$

$$= \sup_{\theta} \mathbb{E}_{p(x,y)}[\theta] - \log \mathbb{E}_{p(x)\otimes p(y)}[e^{\theta}] \tag{12}$$

$$= I(X; Y) \tag{13}$$

where the optimal critic $\theta'^*(x', y') = \theta^*(x, y)$ depends only on the non-noise components. $\qquad\square$

**Corollary 1** ($k$-Sliced MI Invariance and Approximation). *For high-dimensional variables with $d > d_{\max}$, the $k$-sliced MI with padding satisfies:*

1. ***Invariance:** For padded variables $X', Y'$ and random projections $\{P_i\}_{i=1}^{k}$:*

$$I_{k\text{-sliced}}(X'; Y') = \frac{1}{k} \sum_{i=1}^{k} I(P_i X'; P_i Y') = I_{k\text{-sliced}}(X; Y)$$

2. ***Approximation Quality:** Under mild regularity conditions:*

$$|I_{k\text{-sliced}}(X; Y) - I(X; Y)| \leq \frac{C}{\sqrt{k}} \cdot \sqrt{Var_P[I(PX; PY)]}$$

*where $C$ is a universal constant and the variance is over random projections.*

*Proof.* We prove the two parts respectively as follows.

**Part 1:** Follows directly from Proposition 3 applied to each projection.

**Part 2:** By the central limit theorem over independent projections:

$$\sqrt{k}(I_{k\text{-sliced}} - \mathbb{E}_P[I(PX; PY)]) \xrightarrow{d} \mathcal{N}(0, \text{Var}_P[I(PX; PY)])$$

The bias $|\mathbb{E}_P[I(PX; PY)] - I(X; Y)|$ depends on the projection dimension and decreases as more projections capture the dependency structure. $\qquad\square$

**Lemma 1** (Hypernetwork Expressiveness). *The dual-path attention architecture with $L$ layers, hidden dimension $d_h$, and $H$ heads can represent any continuous function from empirical distributions to critic parameters with error:*

$$\epsilon_{\mathcal{H}} \leq C \cdot \left(\frac{1}{L} + \frac{1}{H} + \frac{1}{d_h}\right)$$

*where $C$ depends on the smoothness of the target function.*

*Proof Sketch.* The attention mechanism can approximate any continuous set function by the universal approximation theorem for transformers. The dual-path design separates joint and marginal processing, reducing the approximation complexity. The error bound follows from standard approximation theory with the network capacity scaling with $L \cdot H \cdot d_h$. $\qquad\square$

---

**Algorithm 1** Full Training Sequence Generation Pipeline

---

**Require:** Variable dimensions $d_\mathbf{x}, d_\mathbf{y}$, max components $K_{\max} = 60$, samples $N$, max dim $d_{\max} = 8$

1: Randomly select $K \in \{1, 2, \ldots, K_{\max}\}$ and sample weights $\{\pi_i\}_{i=1}^K$ s.t. $\sum_{i=1}^K \pi_i = 1$
2: Set total dimension $d = d_\mathbf{x} + d_\mathbf{y}$
3: **for** each component $i = 1$ to $K$ **do**
4:     Sample mean $\boldsymbol{\mu}_i \in \mathbb{R}^d$ with elements from Uniform$([-5, 5])$
5:     Select rank $m \in \{1, 2, \ldots, d\}$ and generate $\mathbf{W} \in \mathbb{R}^{d \times m}$ with $W_{ij} \sim \mathcal{N}(0, 1)$
6:     Generate diagonal matrix $\mathbf{D}$ with $D_{ii} \sim$ Uniform$(0, 1)$
7:     Compute $\boldsymbol{\Sigma}'_i = \mathbf{W}\mathbf{W}^T + \mathbf{D}$ and normalize to correlation matrix Corr$_i$: Corr$_{i_{jk}} = \frac{\Sigma'_{i_{jk}}}{\sqrt{\Sigma'_{i_{jj}} \Sigma'_{i_{kk}}}}$
8:     Sample variances $\boldsymbol{\sigma}^2 \in \mathbb{R}^d$ from Uniform$([0.01, 10])$
9:     Set $\boldsymbol{\Sigma}_i = \text{diag}(\boldsymbol{\sigma}) \cdot \text{Corr}_i \cdot \text{diag}(\boldsymbol{\sigma})$
10: **end for**
11: Define GMM: $p(\mathbf{z}) = \sum_{i=1}^K \pi_i \mathcal{N}(\mathbf{z}|\boldsymbol{\mu}_i, \boldsymbol{\Sigma}_i)$
12: Sample $\mathbf{Z} = \{z^1, z^2, ..., z^N\} \sim p(\mathbf{z})$ and partition each $z^j$ into $x^j \in \mathbb{R}^{d_\mathbf{x}}$ and $y^j \in \mathbb{R}^{d_\mathbf{y}}$
13: Organize as sequences $\mathbf{X} = \{x^1, ..., x^N\}$ and $\mathbf{Y} = \{y^1, ..., y^N\}$
14: If needed, pad $\mathbf{X}$ and $\mathbf{Y}$ with $\mathcal{N}(0, 1)$ noise to dimensions $d_{\max}$
15: **return** $(\mathbf{X}, \mathbf{Y})$

---

## A.2 ADDITIONAL DISCUSSIONS OF SYNTHETIC DATA GENERATION

### FULL DATA GENERATION PIPELINE

The following outlines the detailed procedure used to generate synthetic data in our experiments.

- *Mixture components number $K$.* We uniformly sample $K$ from the set $\{1, \cdots, 60\}$.

- *Weights $\pi_i$.* We uniformly sample each $\pi_i$ from the interval $[0, 1]$, then set $\pi_i \leftarrow \pi_i / \sum_{j=1}^K \pi_j$.

- *Mean vector $\boldsymbol{\mu}_i$.* Each element in the mean vector is uniformly sampled from the interval $[-5, 5]$.

- *Correlation matrices.* For the correlation matrices in Gaussian copulas and $t$-copula, we introduce a novel low-rank factorization method for covariance matrix construction that ensures meaningful inter-dimensional correlations. We construct the covariance matrix as $\boldsymbol{\Sigma} = \mathbf{W}\mathbf{W}^T + \mathbf{D}$, where $\mathbf{W} \in \mathbb{R}^{d \times m}$ with rank $m \sim$ Uniform$(\{1, 2, ..., d\})$ has entries $W_{ij} \sim \mathcal{N}(0, 1)$, and $\mathbf{D} = \text{diag}(d_1, ..., d_d)$ with $d_i \sim$ Uniform$(0, 1)$ ensures positive definiteness (Proposition 2). This formulation guarantees that the expected absolute correlation between off-diagonal entries scales as $\mathbb{E}[|\rho_{ij}|] \approx \sqrt{m/(m + 0.5)}$ for $i \neq j$, producing stronger correlations when $m$ is small. By controlling the rank parameter $m$, we systematically vary correlation strength from weak (high rank) to strong (low rank), ensuring the hypernetwork encounters the full spectrum of correlation patterns during training.

- *Degree of freedom $\nu$.* For student-$t$ copula, we randomly sample the degree of freedom $\mu$ as $\nu \sim$ Uniform$([2, 30])$ to vary tail behavior. This exposes the hypernetwork to both short and heavy-tailed dependences.

### ADVANTAGES OF THE PROPOSED COVARIANCE MATRIX GENERATION MECHANISM

In this section, we highlight the advantages of our proposed covariance matrix generation mechanism by comparing it to several commonly used alternatives.

We consider three baseline approaches:

- **Full-rank matrix reparameterization**, where the covariance matrix is constructed as $\mathbf{C} = \mathbf{A}\mathbf{A}^T$, with $\mathbf{A} \in \mathbb{R}^{d \times d}$ being a full-rank matrix whose entries are sampled independently from $\mathcal{N}(0, 1)$.

- **Cholesky decomposition**, where $\mathbf{C} = \mathbf{L}\mathbf{L}^T$, and $\mathbf{L} \in \mathbb{R}^{d \times d}$ is a lower triangular matrix with positive diagonal elements. The diagonal entries of $\mathbf{L}$ are sampled from Uniform$(0, 1)$, and the off-diagonal entries from $\mathcal{N}(0, 1)$.

---

**Algorithm 2** Low-Rank Factorization Method for Covariance Matrix Generation

---

**Require:** Target dimension $d \in \mathbb{N}^+$
**Ensure:** Positive definite covariance matrix $\boldsymbol{\Sigma} \in \mathbb{R}^{d \times d}$
 1: Sample rank parameter $m \sim \text{Uniform}(\{1, 2, \ldots, d\})$
 2: Generate factor matrix $\mathbf{W} \in \mathbb{R}^{d \times m}$ with entries $W_{ij} \overset{\text{i.i.d.}}{\sim} \mathcal{N}(0, 1)$
 3: Generate diagonal matrix $\mathbf{D} = \text{diag}(d_1, \ldots, d_d)$ with $d_i \overset{\text{i.i.d.}}{\sim} \text{Uniform}(0, 1)$
 4: Compute covariance matrix $\boldsymbol{\Sigma} = \mathbf{W}\mathbf{W}^T + \mathbf{D}$
 5: **Optional:** Convert to correlation matrix $\mathbf{R}$ with entries:

$$R_{ij} = \frac{\Sigma_{ij}}{\sqrt{\Sigma_{ii}\Sigma_{jj}}}$$

 6: **Optional:** Rescale to final covariance $\boldsymbol{\Sigma}_{\text{final}} = \text{diag}(\boldsymbol{\sigma})\mathbf{R}\text{diag}(\boldsymbol{\sigma})$, $\sigma_i \sim \text{Uniform}(0.1, \sqrt{10})$
 7: **return** $\boldsymbol{\Sigma}$ (or $\boldsymbol{\Sigma}_{\text{final}}$ if rescaled)

---

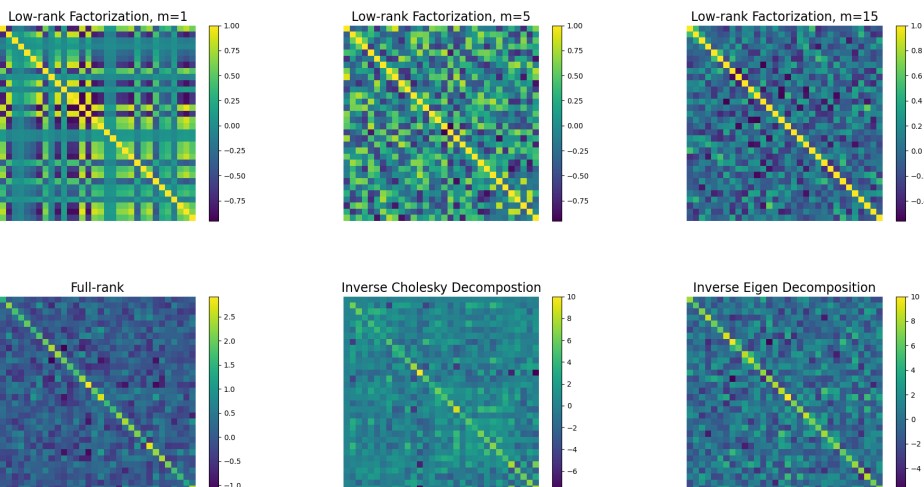

Figure 7: Visualization of correlation matrices generated by various methods. Existing approaches often yield small off-diagonal elements, whereas the low rank factor method adjusts their magnitude by tuning the rank factor $m$.

- **Eigenvalue decomposition**, where $\mathbf{C} = \mathbf{Q}\mathbf{D}\mathbf{Q}^T$, with $\mathbf{D} \in \mathbb{R}^{d \times d}$ being a diagonal matrix with positive entries sampled from $\text{Uniform}[0.1, 10.1)$, and $\mathbf{Q} \in \mathbb{R}^{d \times d}$ being an orthogonal matrix obtained via QR decomposition of a random matrix with entries from $\mathcal{N}(0, 1)$.

While these methods guarantee positive definiteness, they often produce covariance matrices with relatively small off-diagonal entries compared to the diagonal, resulting in limited diversity in the induced dependence structure; see the lower panel in Fig. 7.

In contrast, our method employs a *low-rank factorization* strategy (see Algorithm 1). By tuning the rank parameter $m \leq d$, we can flexibly control the strength of off-diagonal entries, thereby enabling the generation of covariance matrices with highly diverse dependence structures — an important design for ensuring training data diversity. This effect is illustrated in the upper panel of Fig. 7.

### A.3 ADDITIONAL EXPERIMENTAL DETAILS AND RESULTS

EVALUATING INFORMATION BOTTLENECK DYNAMICS WITH *FlashMI*.

Information bottleneck (IB) theory posits that machine learning models compress input data to retain task-relevant information while eliminating irrelevant details (Tishby & Zaslavsky, 2015;

Goldfeld et al., 2019; Shwartz-Ziv & Tishby, 2017; Wongso et al., 2023; Cheng et al., 2019). By monitoring IB dynamics during training, we gain insights into how representations evolve to capture essential dependencies. Specifically, IB theory suggests models minimize the MI between input $\mathbf{x}$ and compressed representation $\mathbf{z}$, denoted $\mathbb{I}(\mathbf{x}, \mathbf{z})$, while maximizing the MI between $\mathbf{z}$ and task labels $\mathbf{y}$, denoted $\mathbb{I}(\mathbf{z}, \mathbf{y})$. This balance optimizes the retention of task-relevant information.

In this experiment, we track the evolution of the IB value during the model training to show how learning serves as a compression mechanism, and in turn showcase the effectiveness of the proposed MI estimator. We train a four-layer MLP (hidden dimension: 512) on the 20 Newsgroups dataset (Lang, 1995), comprising 11,000 text documents across 20 categories. Text data is preprocessed into TF-IDF matrices (Salton et al., 1975) with varying vocabulary sizes (top 1,000, 2,000, or 10,000 terms), with features standardized to zero mean and unit variance. We apply *FlashMI* with 5-sliced MI to compute both $\mathbb{I}(\mathbf{x}, \mathbf{z})$ and $\mathbb{I}(\mathbf{z}, \mathbf{y})$.

Our results demonstrate that MI estimates from *FlashMI*, particularly $\mathbb{I}(\mathbf{z}, \mathbf{y})$, align closely with predictions from IB theory, as illustrated in Fig. 8. Figure 8 presents model training across three vocabulary sizes (1,000, 2,000, and 10,000), with each configuration repeated 10 times to establish reliable means and error bounds (a detailed discussion of overfitting behavior is provided in Appendix ??). Notably, while all models converge to similar training accuracy, they exhibit substantial differences in validation performance. *FlashMI* effectively captures these performance gaps through estimated $\mathbb{I}(\mathbf{z}, \mathbf{y})$ values, with higher MI corresponding to better generalization. This analysis demonstrates how IB theory provides an interpretable framework for understanding model behavior across different data configurations, while simultaneously validating *FlashMI*'s exceptional accuracy and efficiency in estimating MI on complex, high-dimensional real-world data.

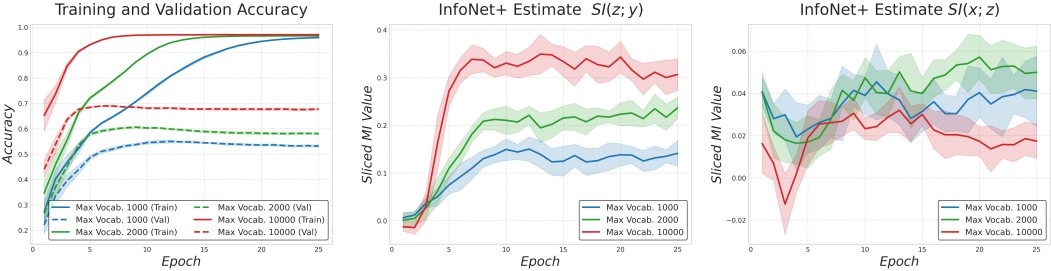

Figure 8: Validation of Information Bottleneck theory on the 20 Newsgroups dataset using *FlashMI* with 5-sliced MI (25 projections). **Left:** Training accuracy (solid lines) converges similarly for all vocabulary sizes, while validation accuracy (dashed lines) shows clear performance differences. **Middle:** *FlashMI* estimated $\mathbb{I}(\mathbf{z}, \mathbf{y})$ precisely tracks these performance differences, with higher values corresponding to better generalization. **Right:** $\mathbb{I}(\mathbf{x}, \mathbf{z})$ remains relatively low across all configurations, as predicted by IB theory. Shaded regions represent error bounds from 10 repeated experiments. These results demonstrate both the interpretability of IB analysis and *FlashMI*'s effectiveness in estimating MI on high-dimensional data.

DETAILS OF INDEPENDENT TESTING EXPERIMENTS

Below are three different relationships between $X$ and $Y$ in high dimensional independence test in sec. 4.2.

(a) **One feature (linear)**: $X, Z \sim \mathcal{N}(0, \mathrm{I}_d)$ i.i.d. and $Y = \frac{1}{\sqrt{2}}\left(\frac{1}{\sqrt{d}}\left(\mathbf{1}^\top X\right)\mathbf{1} + Z\right)$, where $\mathbf{1} :=$ $(1, \ldots, 1)^\top \in \mathbb{R}^d$.

(b) **Two features**: $X, Z \sim \mathcal{N}(0, \mathrm{I}_d)$ i.i.d. and $Y_i = \frac{1}{\sqrt{2}} \begin{cases} \frac{1}{d}\left(\mathbf{1}_{\lfloor d/2 \rfloor}0\ldots0\right)^\top X + Z_i, & i \leq \frac{d}{2} \\ \frac{1}{d}\left(0\ldots0\mathbf{1}_{\lceil d/2 \rceil}\right)^\top X + Z_i, & i > \frac{d}{2}. \end{cases}$

(c) **Independent coordinates**: $X, Z \sim \mathcal{N}(0, \mathrm{I}_d)$ i.i.d. and $Y = \frac{1}{\sqrt{2}}(X + Z)$.

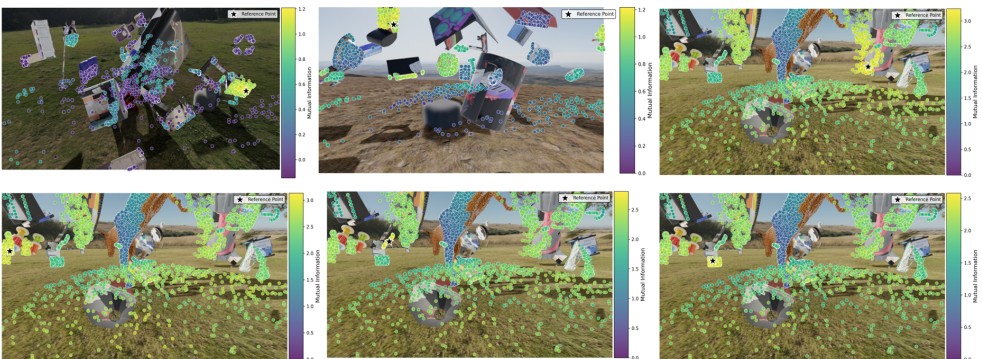

Figure 9: Full visualization results of *FlashMI* estimated motion data.

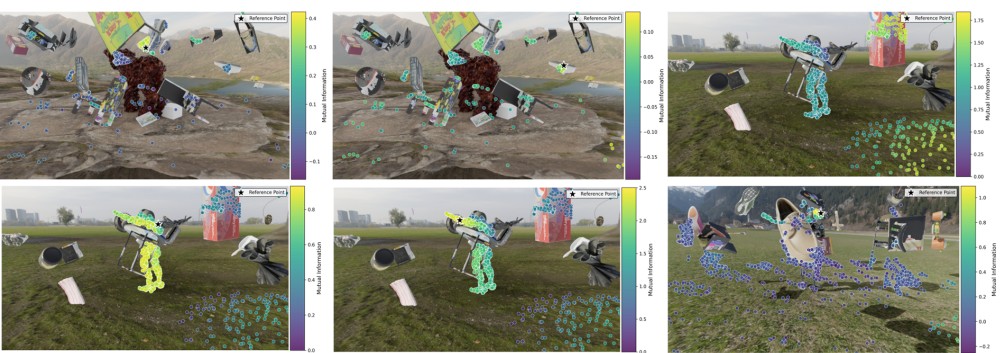

Figure 10: Full visualization results of *FlashMI* estimated motion data.

FULL RESULTS OF VALIDATION ON OUT-OF-DOMAIN MOTION DATA

In this section, we provide detailed results of the experiments on motion data. We only consider points that appear throughout the entire video. Fig. 9 and Fig. 10 show the full visualization results of estimated mutual information between one selected point and other points in the videos.

### A.4 DETAILS OF TRAINING AND ARCHITECTURE

**Neural architecture details of** *FlashMI* For the attention module in *FlashMI*, we configure the dimensionality of the key and value to 1536. The Weight-Decoding MLP comprises seven layers, each of which has 8196 hidden units.

**Optimizer setup** We pre-train *FlashMI* using the Adam optimizer with its default setting over 120,000 iterations, which requires approximately 7 days.

**Batch size and distributional diversity** Each training batch contains 96 independently sampled distributions with 5,000 samples per distribution (so the sequence length fed to the attention module is 5,000), providing the hypernetwork with sufficient statistical evidence to accurately estimate the optimal critic parameters for each distribution type.

**Neural network training protocol**. To ensure a fair comparison, all neural estimators MINE, InfoNCE, MINDE are trained for a maximum of 2,000 epochs with a learning rate of $1 \times 10^{-4}$, employing early stopping if no improvement is observed within 100 epochs. KNIFE is trained with 200 epochs.

**Computational resource**. We pre-train *FlashMI* on a server with 16 NVIDIA H800 GPUs, while all downstream evaluations are conducted using a single H800 GPU and 8-core Intel(R) 8480C CPU.

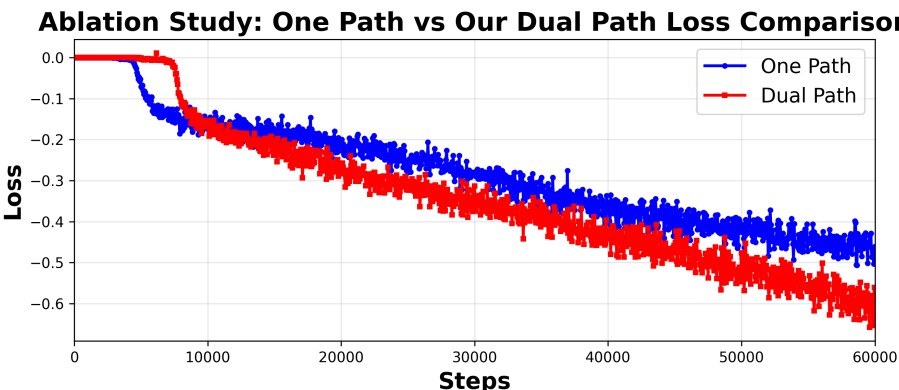

Figure 11: As illustrated in the figure, the training loss of our dual-path model (red line) is significantly lower than that of the only-joint-path model (blue line), under identical hyperparameter settings. This result demonstrates the superiority of the proposed dual-path architecture over single path one.

### A.5 ABLATION STUDY ON DUAL-PATH ARCHITECTURE

In this section, we present an ablation study to evaluate the effectiveness of the proposed dual-path architecture, which is designed to better align with the DV representation. We compare two model variants: both use a maximum dimension of dim = 5, the same learning rate, and an identical joint-path architecture. The only difference is that one model additionally includes a marginal path, while the other does not.

### A.6 DETAILED RESULTS OF INDEPENDENCE TESTING EXPERIMENT

We present additional results for the independence testing experiment described in Paragraph 4.2.

(*Dimensionality trends*) In Fig. 12, we examine the impact of dimensionality on estimation performance by considering three settings with increasing dimensions: 16, 64, and 128. As expected, the test power of our method decreases as dimensionality grows, particularly in small-sample regimes ($n \leq 400$).

(*Visualization of decision making threshold*) In Fig. 13, we illustrate the density distributions of the estimated (sliced) MI values for the dependent and independent groups. The left subplot depicts a scenario in which the two distributions exhibit noticeable overlap, whereas the right subplot illustrates a case with clear separation between them. We also visualize the decision threshold (vertical green line). By adjusting this threshold, one can balance between the false positive rate (FPR) and the false negative rate (FNR) in independence testing.

## B LIMITATIONS

One limitation of *FlashMI* is that while *FlashMI* can effectively handle multivariate data with moderate dimensionalities (e.g. data up to 20 dimensions), it currently requires slicing techniques Goldfeld & Greenewald (2021); Goldfeld et al. (2022) to scale to higher dimensions, where extensive pretraining becomes challenging. Moving forward, we aim to address this limitation by enriching the diversity of the pretraining data, as well as exploring lightweight fine-tuning (e.g., a few gradient steps on the generated critic) to further improve its accuracy in high-dimensional cases.

Another limitation of *FlashMI* is that it may fall short in cases with small sample cases (e.g. $n < 400$), as seen in the independent testing experiments. In such case, the transformer fail to extract informative signals from a small population. That said, our method remains reliable for typical sample sizes encountered in reality (e.g. $n \geq 500$), where our approach consistently matches MINE's accuracy.

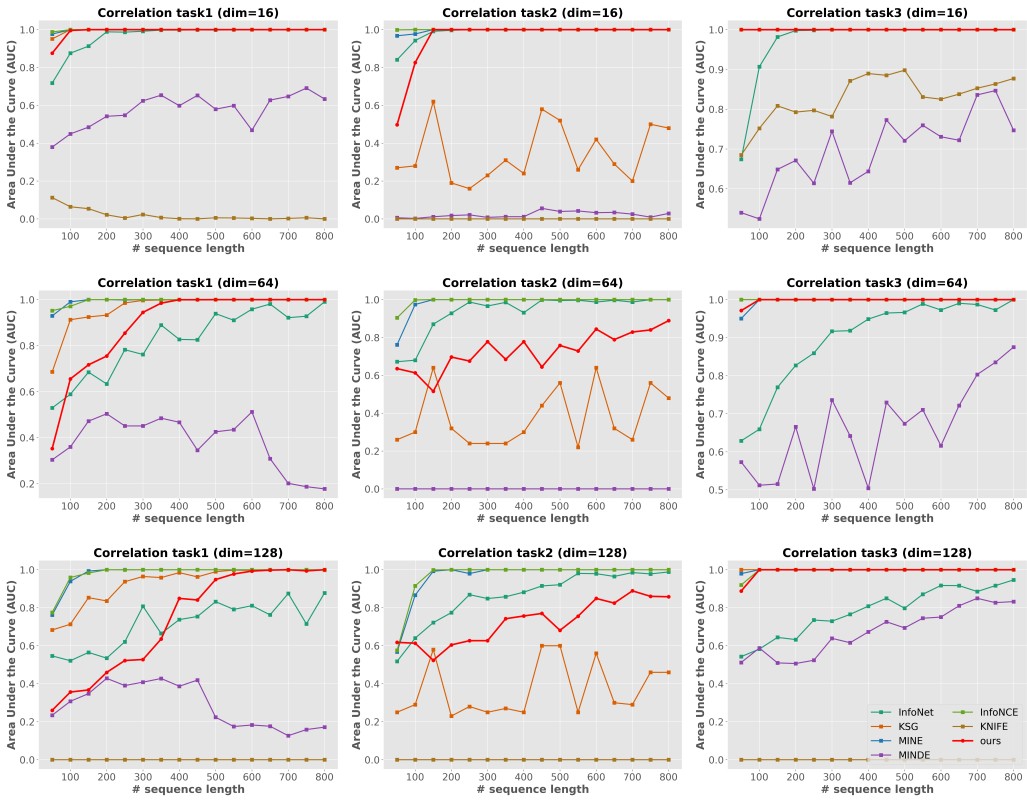

Figure 12: Independence testing across three correlation types and dimensions (16, 64, 128) across seven methods. Each curve plots the ROC-AUC as a function of sequence length $n$. The figure demonstrates that performance degrades with increasing dimensionality.

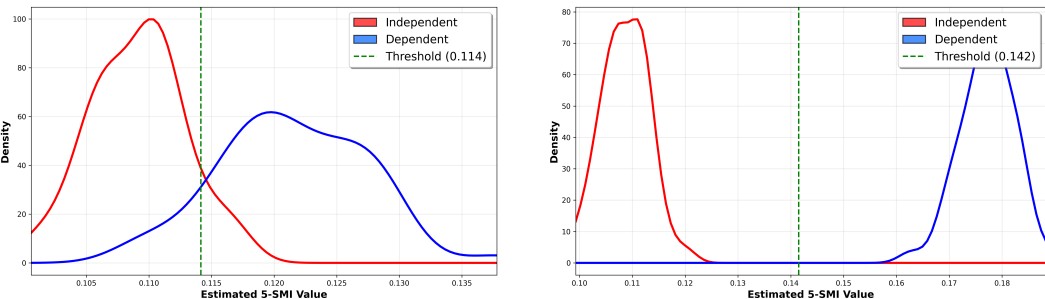

(a) Task 1. Threshold: 0.114; Accuracy: 0.92; Precision: 0.92; Recall: 0.92; F1: 0.92; FP: 4.

(b) Task 3. Threshold: 0.142; Accuracy: 1.00; Precision: 1.00; Recall: 1.00; F1: 1.00; FP: 0.

Figure 13: Visualizing the decision threshold in independence testing experiments. Tests are done with 50 dependent and 50 independent variables, each with a sample size of 250. The figure visualizes the distributions of the 5-SMI estimated with *FlashMI*. By adjusting the threshold, one can flexibly balance between FPR and FRR.

## C   THE USE OF LARGE LANGUAGE MODELS

In the writing of this manuscript, we used large language models solely for grammatical refinement and clarity improvement. The models were not involved in conceptual design, experiment execution, data analysis, or substantive content generation. All scientific contributions, including method design, theoretical derivation, experimental results, and results interpretation are the authors' original work.

