# OpenReview forum: "Neural Mutual Information Estimation in Real Time via Pre-trained Hypernetworks"
_ICLR.cc/2026/Conference — Submitted to ICLR 2026_

### Official Review · Reviewer_M54o · 2025-10-31

**Soundness:** 3
**Presentation:** 3
**Contribution:** 3
**Rating:** 6
**Confidence:** 2

**Summary:**

This paper proposes FlashMI, a pretrained neural architecture for real-time mutual information (MI) estimation. Traditional neural MI estimators (e.g., MINE, InfoNCE) require dataset-specific iterative optimization, which makes them impractical for large-scale or streaming scenarios.
FlashMI reformulates MI estimation as a direct inference task instead of an optimization problem. A hypernetwork generates critic parameters in a single forward pass, enabling efficient estimation.
The model adopts a dual-path architecture (joint/marginal branches) with cross-attention mechanisms consistent with the Donsker–Varadhan formulation. FlashMI is pretrained on large-scale synthetic distributions covering diverse dependency and marginal patterns, enabling zero-shot generalization to unseen data.

**Strengths:**

1.The main contribution is conceptually elegant—the paper redefines MI estimation as a one-step inference problem instead of an optimization task. This paradigm shift offers both theoretical insight and practical value for scalable dependency estimation.
2.he dual-path hypernetwork with cross-attention effectively models relationships between joint and marginal distributions. The inclusion of a noise-padding mechanism enables flexible handling of different input dimensions, enhancing robustness and adaptability.
3.FlashMI achieves substantial computational gains without sacrificing estimation accuracy, making it particularly appealing for real-time and streaming applications.

**Weaknesses:**

1.Limited evaluation on high-dimensional data:
Current experiments focus on low- to mid-dimensional inputs (up to ~20D). The paper mentions potential scaling via slicing or fine-tuning, but provides no empirical evidence or analysis of high-dimensional performance trends.
2.Dependence on synthetic pretraining:
The model’s success heavily depends on the diversity of its synthetic pretraining data. However, the paper does not quantify the variety or coverage of these distributions, making it difficult to assess robustness across real-world domains.

**Questions:**

1.Is the critic parameter generation theoretically guaranteed to approximate the optimal critic, or is it purely empirical?
2.How are the synthetic pretraining distributions designed to capture real-world dependency patterns? Are there metrics to measure their diversity or coverage?

---

> ### Author Response · Authors · 2025-11-21
> **Rebuttal Response (1/2)**
>
> Thank you very much for your thorough review and constructive feedback. Your recognition of method’s **paradigm shift in MI estimation**, the **novelty in architecture design**, and the **substantial efficiency gains** is encouraging. We also value your criticisms regarding **high-dimensional evaluation** and **pretraining data diversity**, which we address below.
>
> &nbsp;
>
> **Rebuttal summary**
> - We report the **high-dimension performance trends** of our method, in addition to the **pre-exist high-dimensional experiments** in the paper.
>
> - Pretraining diversity is supported by **universal approximation guarantees** of our data-generation pipeline, and we introduce a new **Coverage Index** (CI) to quantify this diversity.
>
>
>
> &nbsp;
>
> **W1. Limited evaluation on high-dimensional data; no high-dimensional performance trends**
>
> **A**: Thank you for highlighting this. To explore this trend, **Table 1** compares the performance of a **32D-checkpoint** (which is trained to directly output MI for data up to 32D) with our current **20D-checkpoint**. We observe a clear drop in the 32D model’s accuracy, suggesting that the capacity limit of our method for *directly outputting exact MI* is ~20 dimensions.
>
> **Table 1**. Performance of checkpoints trained for 20D vs. 32D maximum dimensionality on the BMI benchmark. Results on other dimensionality are on the way.
>
> |  | Mn-dense | Spiral | Asinh@Student-t | Uniform |
> | --- | --- | --- | --- | --- |
> | GT | 0.59 | 1.02 | 0.45 | 1.02 |
> | FlashMI-20D (current) | 0.60 | 0.89 | 0.41 | 0.93 |
> | FlashMI-32D  | 0.26 | 0.45 | -0.06 | 0.32 |
>
>
> While our model is less suitable for directly producing *exact* MI values for data beyond 20D, it remains highly effective for high-dimensional dependence measurement *with the aid of slicing*. This is demonstrated by **multiple high-dimensional experiments**, as summarized in **Table 2** below.
>
>
> **Table 2**. A list of the high-dimensional experiments in this work and the data dimensionalities.
>
> | **Task** | **Data dimensionality ($d_\mathbf{x}$, $d_{\textbf{y}}$)** | Sections |
> | --- | --- | --- |
> | High-dimensional independence test | 128, 128 | Section 4.2 |
> | CLIP embedding analysis | 512, 512 | Section 4.2 |
> | Robotics manipulation (**new**) | 100, 100 | Section 4.2 |
> | Information bottleneck analysis of text classifier | 128, 128 | Appendix A.3 |
>
>
>
> &nbsp;
>
> **W2. The model’s success heavily depends on the diversity of its synthetic pretraining data. However, the paper does not quantify the variety of these distributions**
>
> **A**: This is a very reasonable concern. Indeed, pretraining diversity is crucial for our method's performance. We address this issue from two complementary angles.
>
> **1) Ensuring distributional diversity via universal density approximators**. Our synthetic data generator is deliberately designed to span a broad family of distributions:
>
> - The **mixture of Gaussian copulas** used in Eq. (6) is a *universal approximator* for dependence structures given sufficient mixture components [C]. Here, we use up to 60 mixtures, substantially more than the 32 mixtures shown in [C] to achieve accurate dependence approximation.
> - The **normalizing flows** employed in Eq. (7) are *universal approximators* for marginal distributions [D], ensuring rich and varied marginal patterns during pretraining.
>
> Together, our pretraining yield a highly diverse synthetic distribution family, enabling our model to generalize to unseen real-world dependencies, as seen in the experiments.
>
> **2) Quantifying coverage via Coverage Index (CI).** To the best of our knowledge, there does not exist such a metric for measuring “the diversity of distributions” in literature. We therefore introduce a proxy metric $CI$  to implicitly measure such diversity. Formally, $CI \in [0, 1]$ is defined as
>
> $CI := 1 - \mathbb{E}[|\frac{I_\text{GT}-\hat{I}}{I_\text{GT}}|]$
>
> where $\hat{I}$  is our predicted MI and $I_\text{GT}$ is the ground truth MI. The **expectation is taken over the BMI benchmark** [A], which spans diverse, heterogeneous distributions not directly seen during pretraining. If the pretraining distributions exhibit broad coverage, **CI should be close to 1 without any fine-tuning** – precisely what we observe in **Figure 6a**.

---

> ### Author Response · Authors · 2025-11-21
> **Rebuttal Response (2/2)**
>
> &nbsp;
>
> **Q1. Is the critic parameter generation theoretically guaranteed to approximate the optimal critic? Or is it empirical?**
>
> **A**: Yes! Our method comes with a solid theoretical guarantee:
>
> - Proposition 1 in Appendix A shows that our estimator is *consistent* for any dataset $\mathcal{D}$ such that $p(\mathcal{D}) > 0$ for our pretraining distribution $p$. This means that the critic generated by our hypernetwork converges to the optimal critic for any dataset $\mathcal{D}$ within the support of $p$.
> - On the other hand, our pretrain distribution $p$ is designed to cover distributions as diverse as possible (it is an universal density approximator), as already discussed in **W2**. This ensures that  $p(\mathcal{D}) > 0$ holds for most datasets seen at test time.
>
> We have clarified this point in the revised manuscript (see the highlighted texts). Our strong empirical performance across real-world datasets further support the theoretical claim.
>
> &nbsp;
>
> **Q2. How are the synthetic pretraining distributions designed to capture real-world dependency? Any metric for quantifying dependency diversity?**
>
> **A**: In this work, we capture diverse dependencies in pretraining through **a large mixture of Gaussian and $t$-copulas** (Eq.6). Recent advances in vector copula theory [B, C] showed that such mixture is a **universal approximator to the true dependence structure** between $\textbf{x}$  and $\textbf{y}$  given large number of mixtures. While [C] demonstrates state-of-the-art approximation accuracy with 32 mixtures, we use up to 60 in pretraining, which we believe is sufficient to cover a wide range of realistic dependences.
>
> Regarding metric for dependency diversity, please refer to our response to **W2**, where we introduce the **Coverage Index (CI)** as a proxy measure.
>
> &nbsp;
>
> **References**
>
> [A] Czyz et.al. *Beyond normal on the evaluation of mutual information estimators.* NeurIPS *2022*
>
> [B] Fan et.al. *Vector Copulas*. Journal of Econometrics, 2023.
>
> [C] Chen et.al. *Neural Mutual Information Estimation with Vector Copulas*. NeurIPS 2025.
>
> [D] Papamakarios et. al. *Normalizing Flows for Probabilistic Modeling and Inference.* JMLR 2022.

---

> ### Author Response · Authors · 2025-11-28
> **Gental Reminder of Discussion**
>
> Dear Reviewer M54o
>
> Thank you once again for your time and the constructive feedback. We would be very grateful to learn from you whether our rebuttal has addressed your concerns, particularly those regarding **pretraining data diversity**, **high-dimensional trends,** and **convergence.** We are keen to learn from you any further insights you may have, and we look forward to discussing them in more details.
>
> Best, authors.

---

### Official Review · Reviewer_WrB1 · 2025-11-01

**Soundness:** 3
**Presentation:** 4
**Contribution:** 4
**Rating:** 6
**Confidence:** 3

**Summary:**

This paper addresses a core bottleneck of existing neural mutual information (MI) estimators: the need for costly and time-consuming iterative optimization for each new dataset at test time. To overcome this, the authors propose FlashMI, a pre-trained, foundation-model-like architecture. FlashMI reformulates MI estimation from an "optimization problem" to an "inference problem." At its core is a dual-path, attention-based Hypernetwork. This hypernetwork takes the entire dataset (as a sequence of samples) as input and directly generates the optimal parameters for the "critic network" from the Donsker-Varadhan (DV) formulation in a single forward pass. FlashMI is pre-trained on large-scale, diverse synthetic data (covering various distributions and dependency structures), allowing it to learn general distributional patterns. The model flexibly handles varying input dimensions (via noise padding) and sample sizes (via attention). Extensive experiments demonstrate that FlashMI matches state-of-the-art (SOTA) neural estimators in accuracy while achieving over 100x speedup. Furthermore, it successfully generalizes in a zero-shot manner to real-world applications, such as CLIP embedding analysis and motion trajectory modeling, showing its significant potential as a tool for real-time dependency analysis.

**Strengths:**

1.	Exceptional Efficiency: The key strength is the 100x+ speedup by replacing per-dataset optimization with a single forward pass. This makes real-time neural MI estimation practical.
2.	Novel Architecture and Generalization: The dual-path hypernetwork is an innovative design that handles variable inputs. Its effectiveness is proven by the strong zero-shot generalization from synthetic training to real-world tasks (e.g., CLIP, motion data), which is a significant result.
3.	Comprehensive Evaluation: The experimental design is rigorous, testing against optimization-based, pre-training-based, and traditional MI estimators.

**Weaknesses:**

1.	The method still relies on slicing for high-dimensional data (e.g., 512-dim), which is an approximation and limits the core method's direct applicability in such settings.
2.	The model's impressive inference speed comes at the cost of a very high pre-training budget (hardware and time). Its generalization is also entirely dependent on the diversity of the synthetic pre-training data.

**Questions:**

- In the CLIP experiment (512-dim), the caption for Figure 4 mentions "5-sliced MI using 25 random projections". How does this work exactly? Does this mean $k=5$ (as in the k-sliced MI definition) or $S=25$ (the number of projections)?

-  What is the practical value of the max dimension $D$ mentioned on page 6? (Appendix A.1, Alg 1 seems to imply $d_{max}=8$). Please clarify the practical $D$ used and how k-slicing works with it.

- Figure 3 shows that performance (AUC) drops noticeably for small sample sizes (e.g., $n < 400$). How does FlashMI's robustness in this low-sample regime compare to MINE (which can optimize specifically for that small dataset)? Does the hypernetwork need to "see" a sufficient number of samples to accurately infer the distribution's properties?

---

> ### Author Response · Authors · 2025-11-21
> **Rebuttal Response (1/2)**
>
> Thank you for your valuable feedback and for acknowledging our work’s **significance in addressing a core bottleneck of computational efficiency** in existing MI estimators. We deeply appreciate your recognition of our contributions, including **reformulating MI estimation as an inference problem**, the **architectural innovations**, the **strong real-world generalization**, and the **comprehensiveness of evaluation**. In the following, we address your concerns and provide additional information to answer the raised questions.
>
> &nbsp;
>
> **Rebuttal summary**
> - Our method is the **first method** capable of **zero-shot MI inference up to 20D**, and it is **highly useful for high-dimensional dependence analysis** when paried with (k-)slicing.
> - **Pretraining cost** is a **one-time expense**—we will **release checkpoints** so practitioners incur no additional compute.
>
> &nbsp;
>
> **W1: The method still relies on slicing for high-dimensional data (e.g., 512-dim), which is an approximation and limits the core method's direct applicability in such settings.**
>
> **A**: We acknowledge this limitation. That said:
>
> - In many applications, exact MI is often not required; the ***relative strength** or **ordering*** of dependencies is already highly informative (see e.g. the independence testing and the motion-trajectory experiments). Our method is perfectly suitable for providing such signal, particularly when paired with slicing*.
> - Our method is, to the best of our knowledge, the **first** method capable of **computing MI exactly for data** **up to 20 dimensions in a single forward pass**. Existing neural MI estimators cannot perform exact estimation even in these moderate dimensions without substantial optimization overhead.
>
> *Our method has an exceptional affinity to slicing: since we require no optimization, we can pack different slices into a single batch to process them *in parallel*, being highly efficient.
>
> &nbsp;
>
> **W2: The model's impressive inference speed comes at the cost of a very high pre-training budget (hardware and time). Its generalization is also entirely dependent on the diversity of the synthetic pre-training data.**
>
> **A**: This is a valid concern. However, the pre-training cost (~1week with 16x A100) is entirely on the **developer (i.e., the author team) side**. Once pretrained, the community can directly use our publicly released checkpoints (available on HuggingFace/GoogleDrive upon acceptance), just as how one uses Qwen3 or LLama2 in LLM research.
>
> Regarding reliance on pretraining data diversity, we ensure it through a principled generation pipeline based on **universal density approximators** (mixture of Gaussian + flow transformation). Our pretraining use up to 60 mixture components, while prior works [A, B] using far fewer components (e.g., 32) are found sufficient for accurate MI estimation.
>
> &nbsp;
>
> **References**
>
> [A] Butakov et. al. *Mutual Information Estimation with Normalizing Flows*. NeurIPS 2024.
>
> [B] Chen et. al. *Neural Mutual Information Estimation with Vector Copulas*. NeurIPS 2025.

---

> ### Author Response · Authors · 2025-11-21
> **Rebuttal Response (2/2)**
>
> &nbsp;
>
> **Q1: Question regarding CLIP Experiment's k-Sliced MI Setup.**
>
> **A:** Your understanding is fully correct. “5-sliced MI using 25 random projections” means k=5 in the k-sliced MI definition (so we project data onto 5-dimensional spaces) and S=25 projections. We have made this point clear in the manuscript (please see the highlighted texts in the experiment section). Thank you once again for pointing out this unclear point.
>
> &nbsp;
>
> **Q2: Value of Maximum Dimension D**
>
> **A:** The maximal dimension D we can process is D=10 per random variable (thereby 20 dimensions for the joint distribution). We have added this detail to the revised manuscript.
>
> &nbsp;
>
> **Q3: How is FlashMI's robustness in low-sample regime compared to MINE / Does the hypernetwork need to "see" a sufficient number of samples?**
>
> **A:** Thank you for the insightful question. Indeed, our method does not exhibit a clear advantage over MINE in small-sample regimes (e.g. $n<400$) and can even perform worse in extreme cases (e.g. $n\leq 128$) , as shown in Figure 3. We have explicitly highlighted this limitation in the “Limitations” section to provide clearer guidance to practitioners.
>
> That said, the method remains highly reliable for **typical sample sizes** encountered in practice (e.g. $n \geq 500$), where our approach consistently matches MINE’s accuracy. Importantly, as you already noted, the efficiency of our method is 100x faster than MINE or other neural network estimators. We believe this accuracy-efficiency trade is completely worthy.

---

> > ### Comment · Reviewer_WrB1 · 2025-11-27
> >
> > Thanks for the detailed rebuttal and clarifications for addressing my concerns.
> >
> > 1. The clarifications regarding the CLIP experiment setup (k=5,S=25) and the maximum dimension (D=10 per variable) are helpful. I appreciate the updates to the manuscript.
> >
> > 2. Pre-training Cost: I agree that releasing pre-trained checkpoints largely mitigates the concern regarding the high pre-training budget for end-users.
> >
> > 3. Small Sample Performance: I appreciate the authors' honesty regarding the performance drop in small-sample regimes (n < 400) compared to MINE. Explicitly adding this to the "Limitations" section is a responsible move that will greatly guide future practitioners.
> >
> > Given the significant efficiency gains and the solid rebuttal, I am maintaining my positive assessment of this paper.

---

> > > ### Author Response · Authors · 2025-11-28
> > > **Thank you for your feedback and the recognition of our work!**
> > >
> > > Dear Reviewer WrB1
> > >
> > > Thank you very much for your active engagement! We truly appreciate your positive assessment, particuarly regarding our **innovative design** and **exceptional efficiency gain**. We are also highly encouraged that our **rebuttal was solid and addressed your concerns.**
> > >
> > > Once again, thank you for your high-quality feedback and the insightful comments, which have been invaluable in improving our work.
> > >
> > > Best, authors.

---

### Official Review · Reviewer_GWbp · 2025-11-05

**Soundness:** 3
**Presentation:** 3
**Contribution:** 2
**Rating:** 4
**Confidence:** 4

**Summary:**

This paper proposes a new mutual information (MI) estimation method based on neural networks. More specifically, the proposed method aims to compute MI in real time. The main bottleneck in this problem is the time required for MI estimation. To address this issue, the authors propose using a frozen model that is separately trained on synthetic data. After training with synthetic data, mutual information is estimated using the DV representation. Through experiments, the paper shows that the proposed method compares favorably with existing approaches.

The idea of using synthetic data is interesting. However, a similar approach could be implemented simply by extending the MINE method with synthetic data. Moreover, it lacks to comparing to important previous work. Therefore, I feel that the novelty may lie more in the model architecture than in the overall framework itself. This point should be carefully verified.

**Strengths:**

1. The idea of using synthetic data for pretraining would be interesting.
2. The proposed method is much faster than existing methods.

**Weaknesses:**

1. DV based representation learning for mutual information is not new. For example, the following paper has already worked on MI based representation learning.

   Neural Methods for Point-wise Dependency Estimation, NeurIPS 2020.

2. Although the synthetic data pre-training is interesting, the approach is used in the computer vision community. Using it for mutual information is new, but it is not significantly novel.

**Questions:**

1. Regarding independence testing, can the proposed method control the false positive rate?

2. It seems possible to train MINE with synthetic data and then use the trained model to estimate MI, similar to the proposed method. Would the proposed approach still outperform MINE in this setting?

3. Similar to Q4, I feel that the novelty of this paper mainly lies in the proposed model architecture. Could the authors provide an ablation study to support this claim?

---

> ### Author Response · Authors · 2025-11-21
> **Rebuttal Response (1/2)**
>
> Thank you very much for your careful review and constructive feedback. We greatly appreciate your recognition that our method offers substantial efficiency gains. At the same time, we believe there were some **misunderstandings regarding our core contribution and innovation**, which we clarify as below.
>
> &nbsp;
>
>
> **Rebuttal summary**
> - Our contribution lies in the fundamental **optimization → inference shift** in MI estimation via our novel architecture and pretraining, rather than in the use of DV representation.
> - **Existing methods**, including those cited by the reviewer (e.g. MINE), **can NOT realize this ability** at all **even with the aid of synthetic data**.
>
>
> &nbsp;
>
> **W1. DV-based representation learning for mutual information is not new; see e.g. [A]**
>
> **A**: We totally agree, but the use of DV in MI estimation was not intended as our contribution. Rather, **our core contribution lies in making this process zero-shot**, i.e, shifting it from a time-consuming *optimization* problem to an efficient *inference* problem.
>
> - **Existing DV-based methods**, including both MINE and [A], still require training a dedicated critic network $\theta$ from scratch for every dataset $\mathcal{D}$. MI estimation therefore remains a time-costly *optimization* task.
> - **Our method**, by contrast, pretrains these critic networks in advance, then uses an attentive hypernetwork $\mathcal{H}: \mathcal{D} \to \theta$ to ‘select’ the optimal network for a dataset $\mathcal{D}$ in a single forward pass. This way, MI estimation becomes an efficient *inference* task.
>
> [A] Neural Methods for Point-wise Dependency Estimation, NeurIPS 2020
>
>
> **TLDR**: Our novelty is in the "optimization → inference" shift, rather than the use of DV objective itself. [A] is cited as another example that still requires optimization in MI estimation.
>
> &nbsp;
>
>
> **W2. Synthetic data pre-training is interesting, but it is used in the computer vision community. Using it for mutual information is new but not significantly novel.**
>
> **A**:  While the *concept* of pretraining is general, successfully **adapting pretraining to a new task often requires non-trivial technical innovation**. A well-known example is GPT: although pretraining existed in CV long before, it only became effective for texts through **task-specific innovations** e.g.,  attention mechanism and next-token prediction.
>
> Our work follows the same story: making pretraining work for information theory requires our **dual-path attentive hypernetwork architecture**, which is the key to the 'optimization → inference' shift (discussed further in Q2/Q3 below), as well as **large-scale diversity-ensuring synthetic pretraining**, which ensures our method's generalizability to real-world data.
>
> **TLDR**: Pretraining is a general idea, but making it work for MI estimate requires non-trivial, task-specific innovations (neural architecture, pretraining algorithm) — our key contribution.

---

> ### Author Response · Authors · 2025-11-21
> **Rebuttal Response (2/2)**
>
> &nbsp;
>
> **Q1**. **Is it possible to control the false positive rate in the independence test?**
>
> **A**: Yes! Our method offers flexible control over the **false positive rate (FPR)** by **tuning the decision threshold**. For this task, we have used Sliced MI (SMI) as the test statistic, where we declare dependence if SMI > threshold and independence otherwise. By tuning this threshold, one can easily trade FPR with false rejection rate (FRR). Figure 13 provides a visualization.
>
> **TLDR:** The FPR can be easily controlled by adjusting the SMI decision threshold, as visualized in Figure 13 in the appendix.
>
> &nbsp;
>
> **Q2. It seems possible to train MINE with synthetic data and then use the trained model to estimate MI?**
>
> **A**: No, this will not work. To see this, let us look into the details of naive MINE and our method:
>
> - **Naive MINE** is designed to estimate MI for a *single* distribution $p(x, y)$ only. Once trained, the network approximates the log ratio $\log p(x, y)/p(x)p(y)$ up to a constant. If MINE is trained on a synthetic distribution $p_{\text{syn}}(x, y)$, the resultant network will only learn $\log p_{\text{syn}}(x, y)/p_{\text{syn}}(x)p_{\text{syn}}(y)$, which does not transfer to other dataset. So, whenever a new dataset comes, synthetic MINE still has to be retrained to adapt to that dataset.
>
> - **Our work**, on the contrary, learns how to map *different* datasets to the corresponding optimal network (i.e. the density ratio functions) by the use of a hypernetwork. This process can be viewed as that we (pre)train multiple MINE networks in advance simultaneously, each of which is adapted to a specific dataset, and we use a hypernetwork to "route" to the correct MINE network at test time. No retraining is needed.
>
> **TLDR**: MINE + synthetic data only learns the ratio of that synthetic dataset, which does not transfer to a new dataset. We instead learn how to map different dataset to the correct ratio.
>
> &nbsp;
>
> **Q3. The novelty seems mostly in architectural design. Could the author provide an ablation study to support the architecture design?**
>
> **A**: Indeed, architecture design is our central contribution. However, its importance is far beyond what it may initially appear:
>
> - Our **hypernetwork-based architecture** is the **key to realizing the ‘optimization → inference’ shift**, enabling direct mapping of *input dataset* to the *optimal network*. Architectures used in existing MI estimators (e.g. MLP, resnet) maps a single data point to a scalar (the log density ratio at this data point), therefore can not realize this shift at all.
> - Our **noise padding module** and **attention module** enable us to **flexibly handle varying datasets** with different input dimensions and sample sizes with a single, unified model. This allows the network to take any dataset $\mathcal{D}$ as the input, which is crucial for realizing the dataset → underlying ratio function map.
> - The **dual-path** yields **more effective representations for dependence modeling** by separately processing joint and marginal distributions, leveraging the inductive bias that these distributions exhibit different statistical properties. A **new ablation study** is shown in Table 1 below.
>
> **Table 1**. Comparing the training losses of dual-path design vs single-path design. Dual-path design clearly attains lower training loss and converges faster.
>
> |  | 10k | 20k | 30k | 40k | 50k | 60k |
> | --- | --- | --- | --- | --- | --- | --- |
> | **Dual-Path loss ($\downarrow$)**  | -0.165 | -0.2595 | -0.3451 | -0.4348 | -0.5324 | -0.6209 |
> | **Single-Path loss ($\downarrow$)** | -0.1545 | -0.2183 | -0.2803 | -0.3524 | -0.4258 | -0.4738 |
>
> **TLDR:** Our architecture (hypernetwork + noise-padding + attention) is essential for achieving the optimization→inference shift. The dual-path design offers clear empirical advantages.

---

> ### Author Response · Authors · 2025-11-28
> **Gental Reminder of Discussion**
>
> Dear Reviewer GWbp
>
> Thank you once again for your time and effort in reviewing our work. We would be very grateful to learn from you whether our rebuttal has addressed your concerns, particularly those arising from **potential misunderstandings regarding our novelty and core contribution** (both are **acknolwedged by other reviewers**)**.** We understand your time constraints but your insights would be invaluable in strengthening our work.
>
> Best, authors.

---

### Author Response · Authors · 2025-11-25
**Summary of Rebuttal**

We thank the reviewers for their time, effort, and constructive feedback. Reviewers consistently recognized that our method **achieves exceptional efficiency gain in MI estimation** without sacrificing accuracy [GWbp, WrB1, M54o], enabling an **elegant optimization → inference shift** [WrB1, M54o],  and introduces **a novel architecture design tailored for dependency analysis** [GWbp, WrB1, M54o]. Reviewers also highlighted the **zero-shot generation** ability of our method to unseen real-world data [WrB1, M54o] and our **comprehensive evaluation** [WrB1] against optimization-based, pretraining-based and traditional MI estimators. We are enouraged that reviewers found our method **significant for real-time and streaming dependency analysis**.

### *Key concerns/questions addressed*

- **Misunderstanding regarding the novelty and contribution of the proposed framework** [**GWbp**]
    - **Concern**: Reviewer [GWbp] noted that our use of DV representation for MI estimation is not new, and suggested that extending MINE with synthetic data may also work.
    - **Clarification**: Our novelty lies *NOT* in the use of DV itself, but in achieving *zero-shot MI estimation* through our *attentive hypernetwork architecture* and *large-scale synthetic pretraining*.
    - Crucially, this zero-shot ability is fundamentally impossible for existing DV methods (e.g. MINE) *even with synthetic data*, as this strategy will only learn the density ratio for that specific synthetic distribution (which is only useful for estimating the MI of that distribution). Retraining is still required for a new dataset.
    -  Our method, on the contrary, learns to map any set of samples to the corresponding ratio function via a hypernetwork pretrained with diverse synthesized distributions and uses it to directly generate the ratio function for a set of samples at test time.
    - Other reviewers recognized the significance of our contribution e.g. "This paradigm shift offers theoretical insight and practical value for scalable dependency estimation" (M54o).
- **High-dimensional dependency measuement [WrB1, M54o]**
    - **Concern**: Reviewers [WrB1, M54o] pointed out that our method relies on slicing in very high dimensions.
    - **Clarification**: While our method works best with slicing in high dimensions, it nevertheless reliably quantifies statistical dependence up to *1024D* within 2secs, as demonstrated by our high-dimensional experiments. For many applications, exact MI is not needed; the *relative strength* or *ordering* of dependencies is already highly informative.
    - Moreover, our method is still the *first* capable of directly outputting *exact* MI up to 20D without any optimization or slicing.
    - For completeness, we have also reported high-dimensional trends of our method for direct MI estimation in rebuttal.
- **Pretraining data diversity [WrB1, M54o]**.
    - **Concern**: Reviewers [WrB1, M54o] expressed concerns about the diversity of our synthetic pretraining data and enquired about corresponding diversity metric.
    - **Clarification**: Pretraining diversity is ensured through two complementary theoretical guarantees. Mixture of Gaussian copulas (Eq. 6) are *universal approximators for dependence structures* given sufficient mixture components; we use up to 60 mixtures, substantially exceeding the 32 components shown sufficient for state-of-the-art approximation accuracy in recent literature. Normalizing flows (Eq. 7) are *universal approximators for marginal distributions*, ensuring rich and varied marginal patterns during pretraining.
    - A new *Coverage Index* (CI)  $\in [0, 1]$ defined on the BMI benchmark is introduced to quantify distributional coverage. BMI benchmark spans diverse distributions unseen during pretraining. If pretraining distributions exhibit broad coverage, CI should be close to 1 without fine-tuning—precisely what we observe in Fig 6a.

Other concerns, such as pretraining cost, false positive rate control, and experimental clarity are also addressed in rebuttal.
### *Improvements to the manuscript*
During the rebuttal period, we made the following substantive improvements to the manuscript. All revisions are highlighted in red in the updated manuscript.
| Revision | Location | Addresses |
|----------|----------|-----------|
| Robotics manipulation experiment with high-dimensional states | `Experiment 4.2` | WrB1 W1, M54o W1 |
| Ablation study: dual-path vs. single-path architecture | `Appendix A5` | GWbp Q3 |
| Theoretical justification for dependence-structure diversity | `Section 3.2` | WrB1 W2, M54o W2, Q2 |
| Proof of convergence to optimal critic | `Section 3.2 + Appendix A` | M54o Q1 |
| Discussion contrasting FlashMI with additional related works | `Problem Statement` | GWbp |
| Clarified experimental settings (max dimensionality D, k-SMI setups) | `Experiment 4.1` | WrB1 Q2, Q3 |
| Rewritten Limitations section with practitioner guidance | `Appendix B` | WrB1 W1, Q3 |

---

> ### Author Response · Authors · 2025-12-04
> **Reviewer-Centric Synthesis**
>
> **Reviewer GWbp**
>
> Reviewer GWbp's concerns centered on whether the use of DV representation and synthetic pretraining constitutes sufficient novelty, suggesting that MINE with synthetic data might achieve similar results. As detailed above, we clarified that the contribution lies in the zero-shot inference capability enabled by the hypernetwork architecture, not in the use of DV itself.
>
> All three questions were answered: (Q1) FPR can be controlled via threshold tuning; (Q2) MINE + synthetic data cannot achieve zero-shot inference because it learns distribution-specific ratios that do not transfer; (Q3) architecture ablation study now provided in Appendix A5.
>
> We note that Reviewer GWbp acknowledged the efficiency gains ("The proposed method is much faster than existing methods") and found synthetic pretraining "interesting." Reviewer GWbp did not respond after our clarification, and the discussion period ended before further engagement could occur.
>
> **Reviewer WrB1**
>
> Reviewer WrB1 demonstrated strong understanding of our contribution, praising the "100× speedup," "novel architecture and generalization," and "comprehensive evaluation." Their concerns focused on reliance on slicing for high dimensions, pretraining cost, and small-sample performance.
>
> All concerns were addressed: we clarified FlashMI's unique capability for exact MI up to 20D and efficient high-dimensional dependence analysis up to 1024D in under 2 seconds; we confirmed public checkpoint release to eliminate practitioner compute burden; and we documented small-sample limitations in the revised manuscript.
>
> Reviewer WrB1 confirmed in their follow-up comment: **"Thanks for the detailed rebuttal and clarifications for addressing my concerns... Given the significant efficiency gains and the solid rebuttal, I am maintaining my positive assessment of this paper."**
>
> **Reviewer M54o**
>
> Reviewer M54o recognized the "conceptually elegant" paradigm shift and praised the dual-path hypernetwork design. Their concerns focused on limited high-dimensional evaluation trends and pretraining data diversity quantification.
>
> Both concerns were addressed: we added high-dimensional performance trend analysis, introduced the Coverage Index (CI) metric, and provided theoretical justifications via universal approximation guarantees. We also answered their question about theoretical convergence guarantees by referencing Proposition 1 in Section 3.2 and Appendix A1.
>
> Reviewer M54o did not respond after our rebuttal. The concerns they raised have been addressed with new experiments, theoretical justifications, and quantitative metrics.
>
> **Summary**
>
> FlashMI achieves 100× speedup over state-of-the-art neural MI estimators while matching their accuracy. It represents the first method capable of zero-shot MI inference up to 20D, reliably quantifies dependence up to 1024D in under 2 seconds, and demonstrates strong zero-shot generalization to real-world applications including CLIP embeddings, motion trajectories, and robotics manipulation.
>
> Both Reviewers WrB1 and M54o recognized the paradigm shift from optimization to inference as the core contribution and expressed positive assessments. Reviewer GWbp's concerns appear to stem from interpreting our novelty as the use of DV representation rather than the zero-shot inference capability; we have provided detailed clarification on this point. After the rebuttal period, no reviewer raised concerns about technical soundness, and all specific questions have been answered.
>
> We believe the revisions and clarifications provided during the rebuttal period have addressed the reviewers' concerns, and we thank the AC for their time and consideration.
>
> Best regards,
> The Authors

---

### Author Response · Authors · 2025-12-03
**Brief Context Summary for AC Decision**

Dear AC,

Thank you very much for taking care of our submission. We are writing to provide a summary of context that may support an informed decision.

Overall, our submission received mixed initial scores (**664**):

- Both **positive reviewers [WrB1, M54o]** demonstrated a strong understanding of our work and **recognized the novelty and significance of our contribution (the optimization → inference paradigm shift for MI/dependence estimation)**. Reviewer WrB1 also explicitly acknowledged that our rebuttal was solid.
- However, **the only negative reviewer [GWbp]** appears to **misunderstand our key contribution and** **the innovations required for the paradigm shift**. We believe we have fully addressed these misunderstandings in rebuttal; however, due to the abrupt cut of the discussion period, we have no chance to hear back from this reviewer.

We respectfully ask the AC to take this context into account for a fair assessment. Please see the following for a more comprehensive summary and reviewer-centric briefings. Thank you once again for your time and hard work in handling our submission.

---

### Meta-Review · Area_Chair_992c · 2026-01-03

**Summary:**

Reviewer GWbp: The idea of using synthetic data for pretraining would be interesting. and the method is much faster than existing methods. However, DV based representation learning for mutual information is not new. Moreover, although the synthetic data pre-training is interesting, the approach is used in the computer vision community. Using it for mutual information is new, but it is not significantly novel.

Reviewer WrB1: The method has strengths in the exceptional efficiency, novel architecture and generalization, and comprehensive evaluation. However, the method still relies on slicing for high-dimensional data, which is an approximation and limits the core method's direct applicability in such settings. The model's impressive inference speed comes at the cost of a very high pre-training budget (hardware and time). Its generalization is also entirely dependent on the diversity of the synthetic pre-training data.

Reviewer M54o: The method offers conceptual contribution and theoretical insights and practical value for scalable dependency estimations. However, it has some weaknesses. Limited evaluation on high-dimensional data: Current experiments focus on low- to mid-dimensional inputs (up to ~20D). The paper mentions potential scaling via slicing or fine-tuning, but provides no empirical evidence or analysis of high-dimensional performance trends. Dependence on synthetic pretraining: The model’s success heavily depends on the diversity of its synthetic pretraining data. However, the paper does not quantify the variety or coverage of these distributions, making it difficult to assess robustness across real-world domains.

**Reviewer Concerns:**

Based on the rebuttal responses from the Reviewer WrB1, this reviewer thinks that the clarifications regarding the CLIP experiment setup (k=5,S=25) and the maximum dimension (D=10 per variable) are helpful and also agreed that releasing pre-trained checkpoints largely mitigates the concern regarding the high pre-training budget for end-users.

For the remaining reviews, after carefully evaluating the rebuttals, I thinks some of the concerns from reviewer M54o were addressed, such as the high-dimensional trends.

**Reviewer Scores:**

Reviewer GWbp: if the reviewer had been able to participate the discussion, I think the reviewer may keep the original rating unchanged or decrease the rating. Based on the rebuttal, the mutual information issue is not addressed satisfactorily.

 Reviewer M54o: if the reviewer had been able to participate the discussion, I think the reviewer may keep the original rating unchanged.

---

### Decision · Program_Chairs · 2026-01-26

Reject